# Neural clocks and Neuropeptide F/Y regulate circadian gene expression in a peripheral metabolic tissue

Renske Erion[1†], Anna N King[1†], Gang Wu[2], John B Hogenesch[3], Amita Sehgal[1*]

[1]Howard Hughes Medical Institute, University of Pennsylvania, Philadelphia, United States; [2]Department of Systems Pharmacology and Translational Therapeutics, University of Pennsylvania, Philadelphia, United States; [3]Department of Molecular and Cellular Physiology, University of Cincinnati, Cincinnati, United States

**Abstract** Metabolic homeostasis requires coordination between circadian clocks in different tissues. Also, systemic signals appear to be required for some transcriptional rhythms in the mammalian liver and the *Drosophila* fat body. Here we show that free-running oscillations of the fat body clock require clock function in the PDF-positive cells of the fly brain. Interestingly, rhythmic expression of the cytochrome P450 transcripts, *sex-specific enzyme 1 (sxe1)* and *Cyp6a21*, which cycle in the fat body independently of the local clock, depends upon clocks in neurons expressing neuropeptide F (NPF). NPF signaling itself is required to drive cycling of *sxe1* and *Cyp6a21* in the fat body, and its mammalian ortholog, Npy, functions similarly to regulate cycling of cytochrome P450 genes in the mouse liver. These data highlight the importance of neuronal clocks for peripheral rhythms, particularly in a specific detoxification pathway, and identify a novel and conserved role for NPF/Npy in circadian rhythms.

*For correspondence: amita@mail.med.upenn.edu

†These authors contributed equally to this work

Competing interests: The author declares that no competing interests exist.

## Introduction

Circadian clocks constitute an endogenous timekeeping system that synchronizes behavior and physiology to changes in the physical environment, such as day and night, imposed by the 24 hr rotation of the earth (*Zheng and Sehgal, 2012*). A coherent circadian system is composed of a cooperative network of tissue-specific circadian clocks, which temporally coordinate and compartmentalize biochemical processes in the organism (*Wijnen and Young, 2006*). Clock disruption is associated with numerous deleterious health consequences including cancer, cardiovascular disease, and metabolic disorders (*Marcheva et al., 2010*; *Marcheva et al., 2013*; *Turek et al., 2005*).

In the fruit fly, *Drosophila melanogaster*, the neuronal clock network is comprised of roughly 150 circadian neurons, which are grouped based on their anatomical location and function in the brain (*Allada and Chung, 2010*). The lateral neurons include the small and large ventral lateral neurons (LNvs), the dorsal lateral neurons (LN$_d$s) and the lateral posterior neurons (LPNs). The dorsal neurons are divided into three subgroups, dorsal neurons (DN) 1, 2, and 3. The small LNvs (sLNvs) have traditionally been referred to as the central clock because they are necessary and sufficient for rest:activity rhythms under constant conditions (*Grima et al., 2004*; *Stoleru et al., 2004*), but recent studies also indicate an important role for the LN$_d$s (*Guo et al., 2014*). The LNvs express the neuropeptide pigment dispersing factor (PDF), which is important for rest:activity rhythms (*Renn et al., 1999*; *Stoleru et al., 2005*; *Lin et al., 2004*; *Yoshii et al., 2009*) and for the function of circadian clocks in some peripheral tissues (*Myers et al., 2003*; *Krupp et al., 2013*). The LN$_d$s constitute a heterogeneous group of neurons differentiated by the expression of peptides and receptors (*Lee et al., 2006*; *Johard et al., 2009*; *Yao and Shafer, 2014*). Thus far, these peptides, which include

**eLife digest** Many processes in the body follow rhythms that repeat over 24 hours and are synchronized to the cycle of day and night. Our sleep pattern is a well-known example, but others include daily fluctuations in body temperature and the production of several hormones. Internal clocks located in the brain and other organs drive these rhythms by altering the activity of certain genes depending on the time of day.

Animals have specific organs that contain enzymes needed to break down toxic molecules in the body, and the levels of several of these enzymes rise and fall over each 24-hour period. In mammals, these enzymes are found in the liver, but in insects they are found in an organ called the fat body. Here, Erion, King et al. set out to determine the extent to which the internal clock in the brain influences the daily rhythms of these enzymes.

The experiments show that a hormone released by the nervous system is required for the levels of the detoxifying enzymes to change in 24-hour cycles. This hormone – termed Neuropeptide F in fruit flies and Neuropeptide Y in mice – is also known to stimulate both mice and fruit flies to eat. Since toxic molecules often enter the body during feeding, Erion, King et al. speculate that it may be beneficial to link the detoxification process to feeding by using the same mechanism to control both processes. The next step following on from this work would be to find out exactly how neuropeptide F drives the 24-hour rhythms in the fat body and other organs.

Neuropeptide F (NPF), have only been implicated in behavioral rhythms (*He et al., 2013a*; *Hermann et al., 2012*; *Hermann-Luibl et al., 2014*).

Most physiological processes require clocks in peripheral tissues, either exclusively or in addition to brain clocks. For instance, a peripheral clock located in the fat body, a tissue analogous to mammalian liver and adipose tissue (*Arrese and Soulages, 2010*), regulates feeding behavior (*Xu et al., 2008*; *Seay and Thummel, 2011*) and nutrient storage (*Xu et al., 2008*) and drives the rhythmic expression of genes involved in metabolism, detoxification, innate immunity, and reproduction (*Xu et al., 2011*). Molecular clocks in the brain and fat body have different effects on metabolism, suggesting that clocks in these two tissues complement each other to maintain metabolic homeostasis (*Xu et al., 2008*). Such homeostasis requires interaction between organismal clocks, but how this occurs, for example whether neuronal clocks regulate fat body clocks, as they do for some other tissue-specific clocks, is not known. In addition, the fat body clock does not regulate all circadian fat body transcripts. 40% of rhythmically expressed fat body transcripts are unperturbed by the absence of a functional fat body clock (*Xu et al., 2011*), suggesting these genes are controlled by rhythmic external factors, which could include light, food, and/or signals from clocks in other tissues (*Wijnen et al., 2006*). Likewise in the mammalian liver, where circadian gene regulation has been well-studied, cyclic expression of many genes persists when the liver clock is ablated (*Kornmann et al., 2007a*). Brain specific rescue of clock function in $Clock^{\Delta 19}$ animals partially restored liver gene expression rhythms (~40%), albeit with compromised amplitude (*Hughes et al., 2012*). The specific signals that mediate this rescue, however, were not identified, although systemic signals that regulate peripheral clocks have been identified (*Cailotto et al., 2009*; *Kornmann et al., 2007a*; *Reddy et al., 2007*; *Oishi et al., 2005*).

The relative simplicity of fly neuroanatomy and physiology, the vast array of genetic tools, and the conservation of molecular mechanisms with mammals make the fly an ideal organism to dissect complex interactions between physiological systems. In this study, we found that neural clocks regulate circadian gene expression in the fly fat body, a peripheral metabolic tissue. We demonstrate that cycling of the core clock gene, *period* (*per*), requires PDF-expressing cells in constant darkness. Interestingly, however, clocks in the NPF-expressing subset of LN$_d$s, as well as NPF itself, are important for driving rhythmic expression of specific cytochrome P450 genes that cycle independently of the fat body clock. Lastly, we show that Npy, the mouse homolog of NPF, regulates transcriptional circadian output in the mouse liver. Microarray analyses reveal that Npy contributes to the rhythmic expression of hundreds of transcripts in the liver, including a subset of cytochrome P450 genes. In

summary, we identified a conserved role for NPF/Npy neuropeptides in the circadian system in coupling neuronal clocks to transcriptional output in peripheral tissues in flies and mice.

## Results

### The central clock regulates the fat body clock in constant darkness

While some peripheral clocks in *Drosophila* are completely autonomous, e.g. malphigian tubules (*Hege et al., 1997*), others rely upon cell-extrinsic factors, in particular the clock in the brain. For example, PDF-positive LNvs are required for rhythmic expression of clock components in the prothoracic gland, a peripheral tissue that gates rhythmic eclosion (*Myers et al., 2003*). In addition, PDF released by neurons in the abdominal ganglion is necessary to set the phase of the clock in oenocytes (*Krupp et al., 2013*), which regulate sex pheromone production and mating behavior (*Krupp et al., 2008*). We investigated whether clocks in PDF-positive LNvs were necessary for clock function in the abdominal fat body. The molecular clock in *Drosophila* consists of an autoregulatory loop in which the transcription factors, CLOCK (CLK) and CYCLE (CYC), activate expression of the genes *period (per)* and *timeless (tim)* and PER and TIM proteins feedback to inhibit the activity of CLK-CYC (*Zheng and Sehgal, 2012*). To disrupt the molecular clock exclusively in PDF-positive cells, we used the GAL4/UAS system to express a dominant-negative version of the CLK transcription factor, CLKΔ. CLKΔ lacks regions of its DNA-binding domain, preventing it from binding DNA and activating transcription of genes, including components of the molecular clock. However, CLKΔ can still heterodimerize with its partner, CYC, through its protein interaction domain (*Tanoue et al., 2004*). Behavioral assays of *Pdf*-GAL4/UAS-CLKΔ flies showed that a majority of the flies had arrhythmic locomotor activity in constant darkness (DD) (*Figure 1A* and *Table 1*), confirming that CLKΔ expression in the LNvs disrupts circadian rhythms.

To assess functionality of the molecular clock in fat body tissue, we measured transcript levels of the core clock gene *per* in abdominal fat bodies over the course of the day (*Figure 1B*). We found that circadian expression of *per* in the fat body was not altered in flies with a disrupted central clock (*Pdf*-GAL4/UAS-CLKΔ) under a 12 hr light: 12 hr dark (LD) cycle (*Figure 1C*). Unlike mammals, peripheral clocks in *Drosophila* can detect light, which acts as the dominant entrainment signal (*Plautz et al., 1997*; *Oishi et al., 2004*). Therefore, under LD conditions, light may directly synchronize oscillations in *per* transcript levels in fat body cells, masking the effects of ablating the central clock. Consequently, we evaluated *per* rhythms in the absence of light. Since rhythmic gene expression dampens under constant conditions and is undetectable in the fat body by the sixth day of DD (*Xu et al., 2011*), we tested rhythmic expression of *per* on the second day in DD (DD2). *per* levels were rhythmic in the fat body of control flies on DD2. In contrast, flies expressing CLKΔ in the LNvs showed an apparent lack of *per* rhythms in the fat body (*Figure 1D*; see Discussion). This suggests that the clock in PDF-positive LNvs influences the peripheral fat body clock in the absence of external environmental cues.

### Rhythmic expression of fat body transcripts that cycle independently of the local tissue clock requires organismal circadian function

The fat body clock regulates roughly 60% of circadian genes in the fat body; the mechanisms that drive daily cycling of the other 40% of circadian genes in this tissue are unknown (*Xu et al., 2011*). Several potential mechanisms could explain rhythmic gene expression in the absence of the local tissue-specific clock, for example, light, nutrients, or clocks located in other tissues. As noted above, many tissues in *Drosophila* have photoreceptors. Therefore, in addition to entraining clocks to the external environment, LD cycles can drive rhythmic transcription via clock-independent pathways (*Wijnen et al., 2006*). LD cycles can even drive a rhythm of feeding (*Xu et al., 2008*), which could lead to cyclic expression of metabolic genes. Nutrients are known to be strong entrainment signals in peripheral tissues; in fact, rhythmic or restricted feeding, even in the absence of a clock, can drive cyclic expression of several fat body genes (*Xu et al., 2011*). Another possibility is that rhythmic expression of specific fat body transcripts requires a clock in another tissue.

To differentiate between light, nutrient, and clock control, we measured daily expression of genes that cycle independently of the fat body clock in *Clk^{jrk}* mutants. *Clk^{jrk}* mutants lack functional clocks in all tissues due to a premature stop codon that eliminates the CLK activation domain

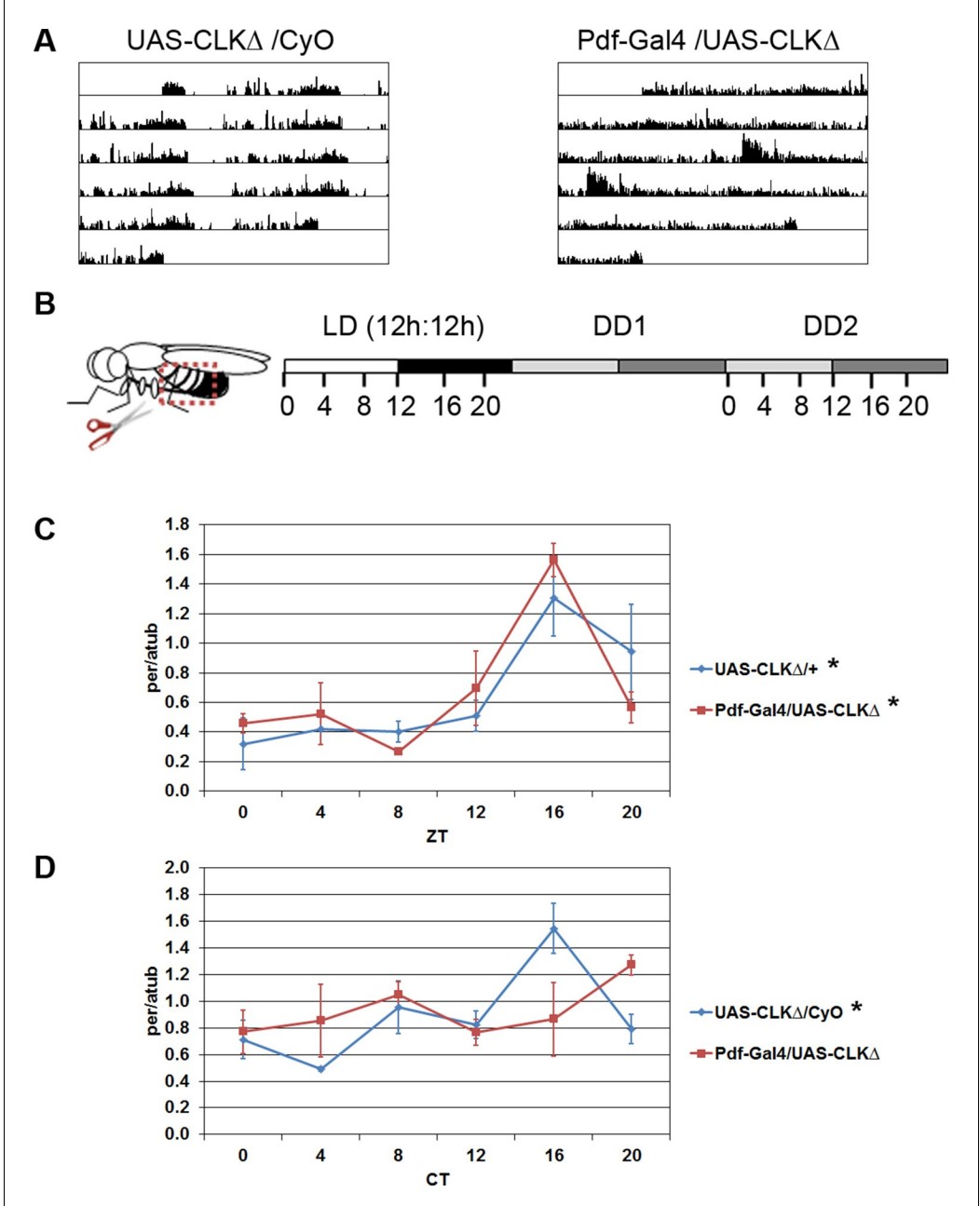

**Figure 1.** Oscillations of *per* in the fat body require an intact central clock in the absence of external cues. (A) Representative double-plotted activity records of individual control UAS-CLKΔ/CyO (left) and *Pdf*-GAL4/UAS-CLKΔ (right) flies over the course of 5 days in constant darkness. (B) Schematic of experimental design. Male flies, aged 7–10 days, were entrained for several days in 12 hr light: 12 hr dark cycles (LD). Male flies were dissected to obtain abdominal fat bodies (dotted red box) either on the last day in LD or on the second day of constant darkness (DD2). Graphs depict mRNA levels, normalized to α–tubulin (atub), over the course of the day in the presence of light (LD; Zeitgeber Time, ZT) or in constant darkness (DD2; Circadian Time, CT). Ablating the central clock (*Pdf*-GAL4/UAS-CLKΔ) (red line) does not affect *per* rhythms in LD (C) but abolishes *per* rhythms in DD2 compared to controls (blue line) (D). Each experiment was repeated independently three times. The average value for each timepoint is plotted with error bars denoting the standard error of the mean (SEM). Significant rhythmicity was determined using JTK_cycle. Asterisk (*) adjacent to genotype label indicates JTK_cycle p value <0.05. See *Table 3* for JTK cycle values.

The following source data is available for figure 1:

**Source data 1.** Data for behavioral analysis and for qPCR analysis of *per* in Pdf-GAL4/UAS-CLKΔ flies.

**Table 1.** Free-running rest:activity rhythms. Clock ablation in *Pdf* neurons (*Pdf*-GAL4/UAS-CLKΔ) or in LNd neurons (*Dvpdf*-GAL4/UAS-CLKΔ; *pdf*-GAL80/+) disrupts free-running behavioral rhythms in flies. Flies with clock ablation in *Npf* neurons (*Npf*-GAL4/UAS-CLKΔ) and *npfr* mutants have normal free-running rhythms.

| Genotype | n | % Rhythmic | Period (hr) | FFT |
|---|---|---|---|---|
| *Pdf*-GAL4/UAS-CLKΔ | 39 | 36 | 23.51 | 0.04 |
| UAS- CLKΔ/CyO | 48 | 90 | 23.71 | 0.06 |
| *Npf*-GAL4/UAS-CLKΔ | 62 | 98 | 24.01 | 0.06 |
| UAS-CLKΔ/+ | 58 | 100 | 23.70 | 0.05 |
| *npfr* | 39 | 95 | 23.66 | 0.05 |
| *npf/+* | 46 | 100 | 23.44 | 0.11 |
| *Dvpdf*-GAL4/UAS-CLKΔ; *pdf*-GAL80/+ | 63 | 68 | 25.51 | 0.06 |
| UAS-CLKΔ/+ | 63 | 100 | 23.91 | 0.08 |

Source data 1. Data for circadian analysis of fly behavior in constant darkness.

(*Allada et al., 1998*). Although $Clk^{jrk}$ mutants cannot sustain feeding rhythms under constant conditions, LD cycles can drive feeding rhythms in $Clk^{jrk}$ flies albeit with a delayed phase relative to wild type flies (*Xu et al., 2008*). We predicted that transcripts driven by light, or even nutrient intake driven by light, would oscillate in $Clk^{jrk}$ mutants in LD with the same or altered phase, while clock-dependent transcripts would not oscillate at all. The genes we tested were selected based on the robustness of their rhythms in the absence of the fat body clock (*Xu et al., 2011*). We found that none of these genes displayed circadian rhythms in $Clk^{jrk}$ mutants, suggesting that although these genes do not require an intact fat body clock, they do require an intact clock in some other tissue (*Figure 2*). In addition to the loss of rhythmic expression in $Clk^{jrk}$ mutants, there were also differences in baseline expression levels. Rhythmic gene expression of *sex-specific enzyme 2 (sxe2)*, a lipase and *CG17562*, an oxidoreductase was eliminated in $Clk^{jrk}$ mutants to produce an intermediate level of gene expression throughout the day (*Figure 2A–B*). In contrast, rhythmic expression as well as overall levels of *sex-specific enzyme 1 (sxe1)*, a cytochrome P450, and *CG14934*, a purported glucosidase involved in glycogen breakdown, were greatly reduced in $Clk^{jrk}$ mutants (*Figure 2C–D*).

## Clocks in NPF-positive neurons drive daily oscillations in expression of fat body transcripts

Having established that circadian expression of genes cycling independently of the fat body clock requires an intact molecular clock elsewhere in the organism, we sought to identity the specific clock population involved. We chose to focus on the regulation of *sxe1* because it has the most robust cycling profile of all the rhythmic fat body clock-independent genes. *sxe1* was named on the basis of its regulation by the sex determination pathway in fly heads and is enriched in the non-neuronal fat body tissue of males (*Fujii and Amrein, 2002*). Early microarray studies looking for cycling transcripts in *Drosophila* heads also indicated that *sxe1* is regulated by the circadian system (*Claridge-Chang et al., 2001*; *McDonald and Rosbash, 2001*; *Ceriani et al., 2002*). However, the nature and function of the circadian control of *sxe1* are unclear. *sxe1* is implicated in xenobiotic detoxification and male courtship behavior (*Fujii et al., 2008*) and may confer cyclic regulation to either or both of these processes.

Rhythms of *sxe1* expression are abolished in $Clk^{jrk}$ mutants in LD, and so we evaluated *sxe1* regulation by other clocks in the presence of light cycles rather than under constant darkness (*Figure 2C*). Our initial discovery that PDF neurons regulate the fat body clock in constant darkness led us to hypothesize that these neurons may also regulate fat body clock-independent genes. Abolishing the clock in PDF cells by expressing CLKΔ under *Pdf*-GAL4, slightly decreased *sxe1* transcript levels in the abdominal fat body, but did not abolish rhythmic expression (*Figure 3A*). This suggests

that although the PDF neurons regulate the fat body clock, these neurons are not the primary drivers of rhythmic *sxe1* expression.

DN1 and $LN_d$ clusters have been implicated in the regulation of circadian behavior (*Zhang et al., 2010*; *Zhang et al., 2010*; *Stoleru et al., 2004*; *Grima et al., 2004*). In fact, DN1s were recently shown to be part of an output circuit regulating rest:activity rhythms (*Cavanaugh et al., 2014*), and clocks in the DN1s are known to mediate other circadian behaviors, such as aspects of the male sex drive rhythm (*Fujii and Amrein 2010*). However, aside from behavioral rhythms, little is known about the functional significance of the DN1 and $LN_d$ clusters in regulating circadian outputs. We investigated whether rhythmic *sxe1* expression requires clocks in the DN1 cluster by using the 911-GAL4 driver to target the DN1s (*Cavanaugh et al., 2014*). Since expressing CLKΔ in the DN1s was lethal, we expressed dominant negative CYCLE, CYCΔ, in the DN1s and found the manipulation did not alter *sxe1* rhythms or expression levels in the fat body (*Figure 3B*).

The six $LN_d$s express NPF (neuropeptide F), sNPF (short neuropeptide F), and ITP (ion transport peptide) in different cells, with some overlap (*Muraro et al., 2013*). In adult males, NPF is expressed in 3 out of 6 $LN_d$s, as well as a subset of the LNvs and some non-clock neurons in the brain (*Lee et al., 2006*; *Hermann et al., 2012*). NPF is also expressed in endocrine cells in the midgut, although the role of NPF in these cells is not known (*Brown et al., 1999*). We first used *Npf*-GAL4 to target the $LN_d$s. Interestingly, we found that expressing CLKΔ under *Npf*-GAL4 severely disrupted expression of *sxe1* (*Figure 3C*). This effect was not specific to the CLKΔ transgene, because *sxe1* expression was also abolished using CYCΔ to disrupt clocks in NPF cells (*Figure 3D*). Since it was possible that expression of CLKΔ or CYCΔ in non-clock NPF cells was disrupting *sxe1* expression, we sought other ways to ablate the clock in $LN_d$ neurons. A subset of the $LN_d$ cluster can also be targeted with the *Dvpdf*-GAL4 driver in combination with *pdf*-GAL80 (*Guo et al., 2014*). Expressing CLKΔ under *Dvpdf*-GAL4;*pdf*-GAL80 reduced *sxe1* levels throughout most of the day, particularly at ZT16, the time of peak *sxe1* expression (*Figure 3E*). The manipulation did not completely abolish rhythmic expression of *sxe1*, presumably because the *Dvpdf*-GAL4 driver does not target all the NPF clock neurons.

We also assessed the circadian expression profile of another fat body clock-independent cytochrome P450 gene, *Cyp6a21*. Fat body expression of *Cyp6a21* robustly cycles in wild type flies but rhythmic expression was dampened in *Npf*-GAL4/UAS-CLKΔ flies, with a relatively small reduction in its overall expression level (*Figure 3F*). This suggests that clocks in NPF-positive neurons have a broader role in regulating the expression of cytochrome P450 genes in the fat body. Furthermore, ablating clocks in NPF-positive cells did not alter rhythmic expression of *per*, indicating that while rhythmic transcriptional output was impaired, the fat body clock remained intact (*Figure 3G*). Together, these data suggest that a subset of $LN_d$s expressing NPF drive rhythmic expression of specific fat body genes.

Next we tested whether overexpressing CLKΔ in NPF-positive neurons in adulthood is sufficient to alter circadian gene expression in the fat body. To limit the expression of CLKΔ to adulthood, we used flies with a tubulin-GAL80^ts transgene (*tub*-GAL80^ts) in addition to *Npf*-GAL4 and CLKΔ transgenes. Tub-GAL80^ts ubiquitously expresses a temperature-sensitive GAL80 protein, which represses GAL4 activity at the permissive temperature of 18°C (*McGuire et al., 2003*). All *Npf*-GAL4/UAS-CLKΔ; *tub*-GAL80^ts/+ flies were raised at 18°C and upon reaching adulthood, control flies were kept at 18°C, while experimental flies were shifted to the restrictive temperature (30°C) to induce CLKΔ expression. We found that after shifting flies to 30°C, expression of *sxe1* remained rhythmic and similar to 18°C controls, suggesting adult-specific clock ablation in NPF-positive neurons is either incomplete or insufficient to affect *sxe1* rhythms (*Figure 3H*). However, this manipulation had a different effect on cyclic expression of *Cyp6a21*. Robust cycling of *Cyp6a21* cycling was maintained in control flies kept at the permissive temperature, although the phase was shifted, perhaps due to the different temperature (18°C) required for this assay. Importantly though, rhythmic expression of *Cyp6a21* was dampened by adult-specific clock ablation in $LN_d$ neurons (*Figure 3I*). Together these data indicate that clocks in NPF-expressing neurons have differential effects on the expression of cycling fat body genes.

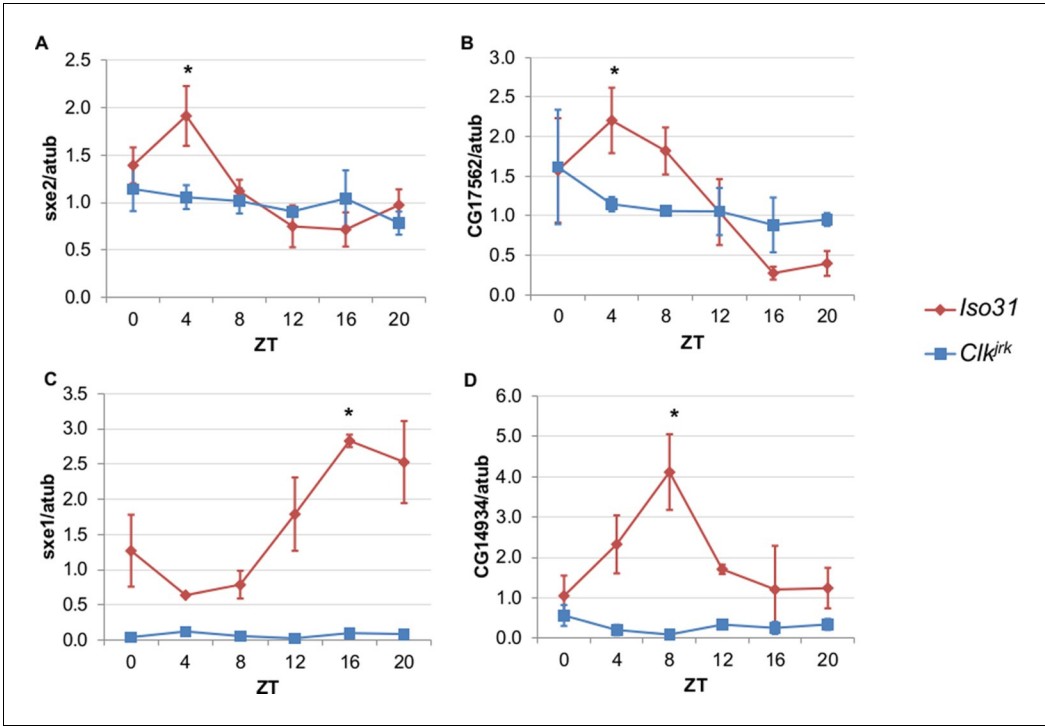

**Figure 2.** Rhythmic expression of genes that cycle independently of the fat body clock requires clocks in other tissues. Daily oscillations of several fat body clock-independent genes were tested in male mutants lacking functional clocks in all tissues, *Clk^irk* mutants, in LD. Rhythmicity of *sxe2* (**A**), *CG17562* (**B**), *sxe1* (**C**), and *CG14934* (**D**) is abolished in *Clk^irk* mutants but is intact in Iso31 wild type controls. All genes were normalized to α−tubulin (atub) levels. Each experiment was repeated independently three times. The average value for each timepoint is plotted with error bars denoting SEM. JTK_cycle p value <0.05 is indicated by an asterisk (*) at the time of peak expression. See *Table 3* for JTK_cycle p values. ZT- Zeitgeber Time.

The following source data is available for figure 2:

**Source data 1.** Data for qPCR analysis of fat body clock-independent genes in Clk^irk mutants.

## NPF-NPF receptor axis regulates rhythmic expression of *sxe1* and *Cyp6a21*

After identifying NPF-positive clock neurons as relevant for rhythmic gene expression in the fat body, we reasoned NPF itself might act as a circadian signal. Indeed, NPF was reported to cycle in a subset of NPF-positive neurons, including LN$_d$s and LNvs (*He et al., 2013a*). NPF regulates a variety of behavioral processes in *Drosophila* including feeding (*Wu et al., 2003*; *2005*; *Lingo et al., 2007*; *Itskov and Ribeiro, 2013*), courtship (*Kim et al., 2013*), aggression (*Dierick and Greenspan, 2007*), and sleep (*He et al., 2013b*). Therefore, we asked if molecular clocks in NPF-positive neurons mediate free-running behavioral rhythms. We found that flies expressing CLKΔ with *Npf*-GAL4 as well as flies carrying a null mutation in *nfpr*, the gene encoding the receptor for NPF, display normal rhythms of rest:activity (*Table 1*). In contrast, *Dvpdf*-GAL4;*pdf*-GAL80 driving UAS-CLKΔ increased the number of arrhythmic flies and slightly lengthened the period of rhythmic flies, further indicating that *Dvpdf*-GAL4;*pdf*-GAL80 and *Npf*-GAL4 do not represent the exact same population of LN$_d$s (*Table 1*). These data suggest that NPF plays at best a minor role in regulating rhythmic locomotor behavior. However, NPF might play a role in other aspects of circadian rhythms, such as circadian control of energy homeostasis.

To determine whether NPF drives rhythmic *sxe1* expression in the fat body, we began by knocking down *npf* in all NPF-positive cells with RNA interference (RNAi), as *npf* mutants are not available. Driving UAS-*npf* RNAi under *Npf*-GAL4 resulted in dampened but still rhythmic *sxe1* expression (*Figure 4A*). Although this manipulation vastly reduced *npf* levels in fly heads (*Figure 4B*), it is

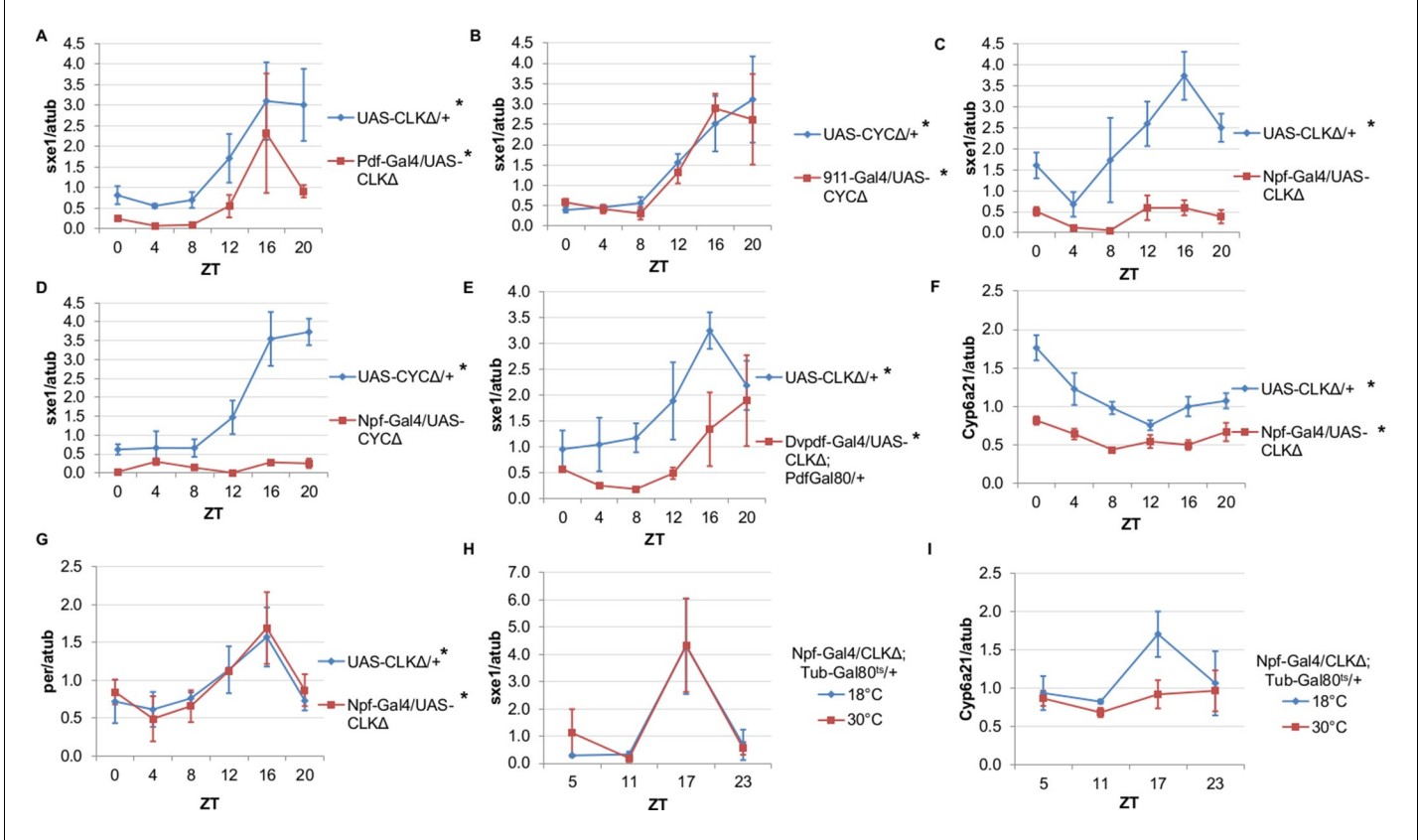

**Figure 3.** NPF-expressing clock neurons regulate rhythmic expression of fat body genes, *sxe1* and *Cyp6a21*. (**A, B**) Ablating the molecular clock by expressing CLKΔ or CYCΔ in either the LNvs (*Pdf*-GAL4) (**A**) or DN1s (911-GAL4) (**B**) does not eliminate rhythmic *sxe1* expression in the fat body. (**C, D**) Expressing CLKΔ (**C**) or CYCΔ (**D**) using *Npf*-GAL4 abolishes rhythmic *sxe1* expression in the fat body. (**E**) Expressing CLKΔ in a subset of LNds (*Dvpdf*-GAL4;*Pdf*-GAL80) also does not eliminate cycling but reduces *sxe1* expression in the fat body. (**F**) *Npf*-GAL4>UAS-CLKΔ abolishes rhythmic *Cyp6a21* expression in the fat body. (**G**) *per* expression is rhythmic in flies expressing UAS-CLKΔ under *Npf*-GAL4. (**H, I**) CLKΔ expression in NPF cells is restricted to adulthood using *Tub*-GAL80ts. (**H**) *sxe1* expression is not affected with adult-specific clock ablation in NPF cells. (**I**) Rhythmic *Cyp6a21* expression is affected in the fat body when *Npf*-GAL4>UAS-CLKΔ expression is induced in adult at 30°C. Each experiment was repeated independently at least twice. The average value for each timepoint is plotted with error bars denoting SEM. JTK_cycle p value <0.05 is indicated by an asterisk (*) next to the genotype label. See *Table 3* for JTK_cycle p values. ZT- Zeitgeber Time.

The following source data is available for figure 3:

**Source data 1.** Data for qPCR analysis of fat body clock-independent genes and clock genes in flies with ablated clock neurons.

possible that very small amounts of NPF can drive some level of cycling; alternatively, knockdown efficiency may have been limited in the NPF-positive clock cells. Thus we tested the null mutant of the sole NPF receptor in *Drosophila*, *npfr* (*Garczynski et al., 2002*). Our results show *sxe1* levels do not cycle and are dramatically reduced in *npfr* mutants, which phenocopies the daily *sxe1* expression profile of flies expressing either CLKΔ or CYCΔ under *Npf*-GAL4 (*Figure 4C*). Rhythmic expression of *Cyp6a21* was also lost in the fat body of *npfr* mutants (*Figure 4D*). We speculate that expressing CLKΔ under *Npf*-GAL4 alters the circadian production or release of NPF. Indeed, mRNA analysis of *Npf*-GAL4/UAS-CLKΔ heads showed that *npf* levels were reduced compared to controls (*Figure 4E*) while cyclic *per* expression, which arises from clock function in many different cells, was unaffected (*Figure 4F*). This result is consistent with reports of loss of NPF expression in LNds of *Clk*irk brains (*Lee et al., 2006*). Taken together, these data suggest circadian clocks in NPF-positive cells regulate NPF expression to subsequently drive *sxe1* and *Cyp6a21* rhythms in the fat body.

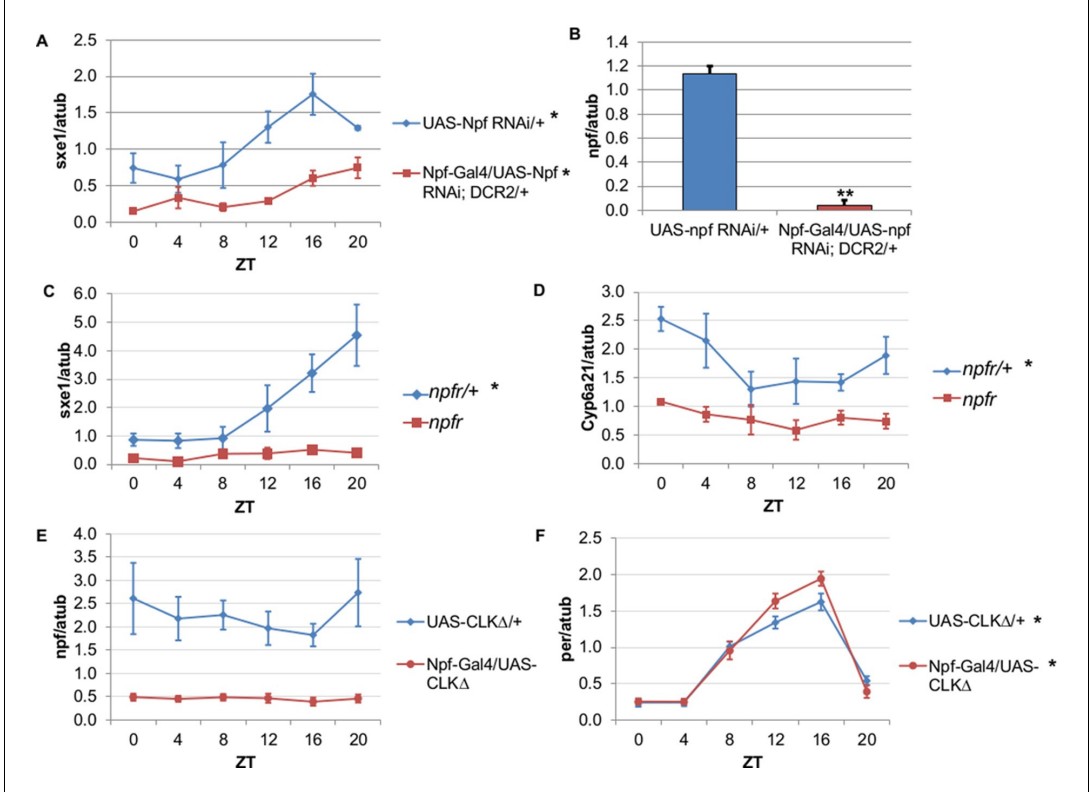

**Figure 4.** NPF is a critical circadian signal for *sxe1* and *Cyp6a21* rhythms in the fat body. (**A**) Knockdown of *npf* in all NPF-positive cells does not eliminate rhythmicity but reduces expression of *sxe1* in the fat body at all times. (**B**) Analysis of *npf* knockdown efficiency in heads of *Npf*-GAL4/UAS-*npf* RNAi; DCR2 (UAS-Dicer2) flies showed a significant reduction in *npf* levels by Student's t-test (**p<0.001). (**C, D**) *sxe1* and *Cyp6a21* expression in the fat body are reduced and do not cycle in homozygous *npfr* mutants compared to heterozygous controls. (**E**) *npf* levels in the heads of *Npf*-GAL4/UAS-CLKΔ are reduced compared to controls (UAS-CLKΔ/+). (**F**) Total *per* levels are not altered in the heads of *Npf*-GAL4/UAS-CLKΔ compared to controls. Each experiment was repeated independently three times except for (**B**) which n=6 for each genotype. The average value ± SEM for each timepoint is plotted. JTK_cycle p value <0.05 is indicated by an asterisk (*) next to the genotype label. See *Table 3* for JTK_cycle p values. ZT- Zeitgeber Time.

The following source data is available for figure 4:

**Source data 1.** Data for qPCR analysis of fat body clock-independent genes and clock genes in flies with perturbed NPF signaling.

## Npy regulates circadian expression of cytochrome P450 genes in the mammalian liver

In mammals, liver-specific circadian clocks play an important role in liver physiology via contributions to glucose homeostasis and xenobiotic clearance (*Gachon et al., 2006*; *Lamia et al., 2008*). Liver clock ablation in mice resembles fat body clock ablation in flies; in particular, ablating liver clocks eliminates rhythmic expression of most, but not all, circadian liver transcripts (*Kornmann et al., 2007a*). Furthermore, rescuing clock function specifically in the brains of *Clock*[Δ19] mutant mice restores rhythmic expression of roughly 40% of circadian liver transcripts (*Hughes et al., 2012*). These data suggest that some circadian transcripts in the liver are driven by systemic signals, perhaps emanating from the master pacemaker in the suprachiasmatic nuclei (SCN) of the hypothalamus (*Mohawk et al., 2012*).

Since we identified NPF in the regulation of circadian gene expression in the fly fat body, we reasoned that the mammalian homolog, Npy, might regulate circadian gene expression in the liver. Thus, we isolated RNA from the livers of male *Npy* knockout (*Npy* KO) mutant mice and wild type controls over the course of an entire day. Although there is no direct mammalian homologue of *sxe1*, we noticed that a similar P450 enzyme involved in xenobiotic detoxification, *Cyp2b10*, also continues to cycle in animals lacking functional liver clocks (*Kornmann et al., 2007a*). We measured *Cyp2b10* levels in *Npy* KO and wild type mice and found that *Cyp2b10* transcript levels did not

display a circadian rhythm in *Npy* KOs (*Figure 5A*). However, circadian expression of the core clock gene *Rev-erb alpha* was unaffected in the livers of *Npy* KOs confirming that the liver clock is still intact (*Figure 5B*). We wondered whether other enzymes involved in xenobiotic detoxification are also regulated by Npy. *Aminolevulinic acid synthase 1 (Alas1)*, is required for P450 synthesis (*Furuyama et al., 2007*) and was also reported to cycle in the absence of the liver clock (*Kornmann et al., 2007a*). Unlike *Cyp2b10*, circadian expression of *Alas1* was unaffected in *Npy* KOs, suggesting that NPY does not regulate global rhythmic detoxification in the liver (*Figure 5C*).

To determine the extent to which loss of Npy impacts gene expression in the liver, we performed genome-wide expression analysis on wild type control and *Npy* KO livers collected at 4 hr intervals over a day in LD. Using the newly developed MetaCycle package (see Materials and methods) and a stringent *P*-value cutoff of $p<0.01$ to detect cyclic transcripts, we found that 289 transcripts were cyclic in controls but not in *Npy* KO, indicating that the oscillation of these transcripts is under the regulation of Npy signaling (*Figure 5D* and *Supplementary file 1*). Furthermore, the loss of transcript cycling was generally not accompanied by differences in expression levels; in other words, the median transcript abundance in wild type animals correlated with that in *Npy* KO (*Figure 5E*). Based on our *Drosophila* data and also the fact that Npy regulates *Cyp2b10* expression, we speculated that Npy might have a broader role in regulating cytochrome P450 gene expression. We examined the microarray data for cyclic P450 transcripts and found several of these genes were not cyclic in *Npy* KOs. Notably, the microarray data confirmed our qPCR data for *Cyp2b10* and indicated that *Cyp2r1, Cyp17a1*, and *Cyp2c70* transcripts also cycle in wild type but not in *Npy* KO liver. In contrast, *Cyp3a13 and Cyp7a1* transcripts cycle robustly in both genotypes.

Lastly, we compared our *Npy* KO data to the previously reported set of liver transcripts whose expression oscillates independently of the liver clock (*Kornmann et al., 2007a*; *2007b*). Among that set, we discovered nine additional liver clock-independent transcripts—*Rbl2, Ddx46, Cirbp, Sqle, Ldb1, Actg1, Hmgcs1, Heca*, and *Ctgf*—that require Npy for robust rhythmic expression (*Table 2*). As only a subset of liver-clock independent transcripts requires Npy for oscillations, other mechanisms likely contribute to rhythmic expression of these genes (further discussed below). Although many genes, including clock genes, continued to cycle in *Npy* knockout livers, the overall phases and amplitudes of expression for cycling transcripts in *Npy* KO slightly differed from those in wild type (*Figure 5—figure supplement 1*). Overall we found that diverse liver circadian transcripts, including cytochrome P450 genes, are influenced by Npy signaling. This report is the first to describe a role for Npy in the circadian regulation of peripheral gene expression in mammals.

## Discussion

In this report we dissect the role of neural clocks in the regulation of circadian gene expression in a peripheral tissue. We find that clocks in PDF-positive neurons influence cycling of the *per* clock gene in the *Drosophila* fat body in the absence of external cues. More importantly, we identify the non-cell autonomous mechanism that underlies cycling of specific fat body transcripts in *Drosophila* and specific liver transcripts in mice. We show that clocks in *Drosophila* NPF-positive neurons drive daily expression of *sxe1* and *Cyp6a21*, fat body genes not controlled by the fat body clock. Likewise, mammalian Npy drives rhythmic expression of specific liver transcripts, indicating a conserved role of NPF/Npy in the control of peripheral circadian rhythms.

Prior to this report, it was proposed that clocks in the brain and fat body interact, but the extent of the interaction and the mechanisms driving it were not identified (*Xu et al., 2008*). Our data suggest that in light:dark cycles, the central clock is not required for cycling of the fat body clock, although we cannot exclude an effect on the phase of cycling. However, in constant conditions, the clock in PDF cells influences the fat body clock, as it does the prothoracic gland clock. Future work will determine whether PDF release into the hemolymph influences the fat body clock (*Talsma et al., 2012*; *Krupp et al., 2013*). Why the central clock regulates only some peripheral clocks in the fly is unclear. Unlike other peripheral clocks, the fat body clock modulates behavioral rhythms, specifically the phase of feeding rhythms, in addition to its own physiology (*Xu et al., 2008*; *2011*; *Seay and Thummel, 2011*). Thus, synchrony between clocks in the brain and fat body is likely essential for metabolic homeostasis.

The circadian system controls behavior and physiology in large part through its regulation of circadian gene expression (*Zheng and Sehgal, 2012*). Tissue-specific gene expression patterns are

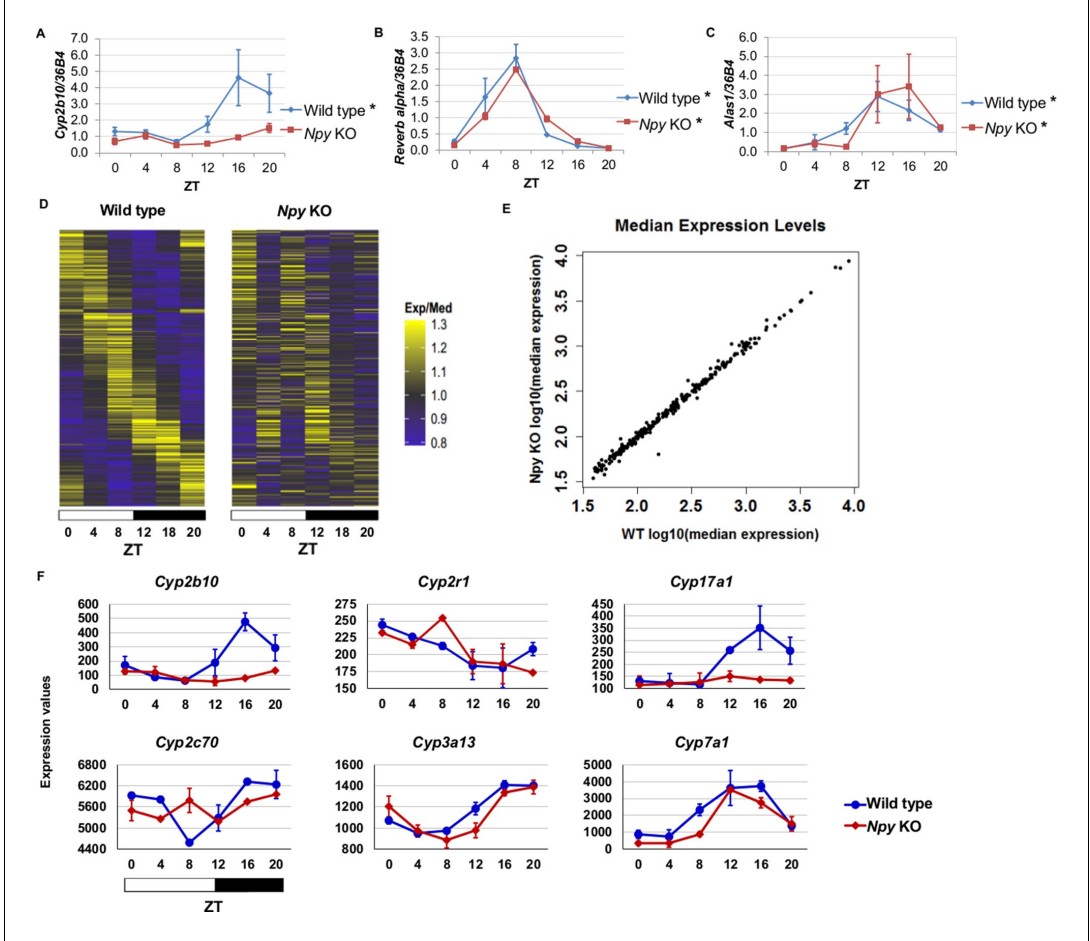

**Figure 5.** Npy regulates circadian expression of cytochrome P450 genes in the murine liver. (A–C) Quantitative PCR analysis in murine livers. Daily oscillations of *Cyp2b10* expression (A) are abolished in *Npy* KOs compared to their background controls (wild type), while oscillations of the circadian gene, *Reverb alpha* (B), are unaffected. (C) Levels of another liver clock-independent gene, *Alas1*, are similar in wild type and *Npy* KO, suggesting Npy does not regulate its rhythmicity. For qPCR data, *n*=3–4 mice for each genotype and time point. Transcript levels were normalized to the housekeeping gene *36B4*. (D–F) Microarray analysis was used to detect transcript expression in livers of *Npy* KO and their background controls collected over the course of 24 hr in LD. (D) The heatmap includes transcripts that oscillate in wild type but not in *Npy* KO liver. Data represent the average transcript abundance from *n*=2 samples for each genotype and timepoint. Here, the MetaCycle *p*-value cutoff of p<0.01 was used to identify cyclic transcripts; p>0.8 was considered not cyclic. (E) The median expression values of the wild type-only cyclic transcripts are not different between *Npy* KO and wild type. (F) Daily expression values of cytochrome P450 genes from microarrays. Cytochrome P450 genes *Cyp2b10*, *Cyp2r1*, *Cyp17a1*, and *Cyp2c70* are cylic in wild type liver but are not cyclic in *Npy* KO liver. *Cyp3a13* and *Cyp7a1* cycle robustly in both wild type and *Npy* KO. Graphs show average ± SEM. ZT- Zeitgeber Time.

The following source data and figure supplement are available for figure 5:

**Source data 1.** Data for qPCR and microarray analysis in wild type and *Npy* knockout mouse liver.

**Figure supplement 1.** MetaCycle analysis of cycling liver transcripts in wild type and *NPY* KO.

thought to be generated primarily by local clocks; however, few studies have comprehensively evaluated rhythmic expression driven by local clocks versus external factors. A previous comparison of gene expression profiles of flies containing or lacking an intact fat body clock found that the fat body clock only regulates ~60% of all circadian fat body genes (*Xu et al., 2011*). Here we report that at least some of the other 40% of circadian fat body genes are regulated by clocks located in other tissues. We found that disrupting clocks in NPF-positive cells abolished rhythmic expression of two cytochrome P450 genes, *sxe1* and *Cyp6a21*. Since we specifically disrupted the molecular clock by expressing CLKΔ or CYCΔ, only NPF-positive cells containing circadian clock components should

**Table 2.** MetaCycle statistics for cycling of liver clock-independent genes in wild type and *Npy* knockout mice. Our *Npy* KO data compared to the previously reported set of liver transcripts whose expression oscillates independently of the liver clock (**Kornmann et al., 2007b**). Ten liver clock-independent transcripts require Npy for robust rhythmic expression.

**Liver clock-independent genes with disrupted cycling of expression in *Npy* KO**

| Affymetrix transcript ID | Gene | WT MetaCycle P-value | WT Phase | WT Median Exp | WT Relative Amp | KO MetaCycle P-value | KO Phase | KO Median Exp | KO Relative Amp |
|---|---|---|---|---|---|---|---|---|---|
| 17503756 | *Rbl2* | 0.0003 | 8.94 | 367.0 | 0.217 | 0.0753 | 9.20 | 338.7 | 0.132 |
| 17287733 | *Ddx46* | 0.0007 | 7.77 | 264.2 | 0.229 | 0.1756 | 11.46 | 262.8 | 0.158 |
| 17235227 | *Cirbp* | 0.0037 | 6.28 | 58.1 | 0.188 | 0.0924 | 7.83 | 54.3 | 0.056 |
| 17311807 | *Sqle* | 0.0039 | 21.00 | 193.4 | 0.790 | 0.7685 | 9.43 | 229.9 | 0.310 |
| 17365314 | *Ldb1* | 0.0067 | 10.38 | 234.1 | 0.164 | 0.1503 | 12.05 | 216.2 | 0.171 |
| 17475360 | *Cyp2b10* | 0.0102 | 17.75 | 155.9 | 0.869 | 0.0782 | 0.00 | 85.7 | 0.486 |
| 17331429 | *Actg1* | 0.0153 | 18.56 | 155.7 | 0.470 | 0.3570 | 1.52 | 124.5 | 0.512 |
| 17290173 | *Hmgcs1* | 0.0230 | 0.50 | 751.5 | 0.687 | 0.1625 | 4.83 | 695.2 | 0.083 |
| 17239493 | *Heca* | 0.0265 | 9.30 | 255.6 | 0.132 | 0.1921 | 10.93 | 258.6 | 0.086 |
| 17232235 | *Ctgf* | 0.0331 | 13.18 | 37.9 | 0.364 | 0.9984 | 11.52 | 42.8 | 0.152 |

**Liver clock-independent gene with altered phase of expression in *Npy* KO**

| Affymetrix transcript ID | Gene | WT MetaCycle P-value | WT Phase | WT Median Exp | WT Relative Amp | KO MetaCycle P-value | KO Phase | KO Median Exp | KO Relative Amp |
|---|---|---|---|---|---|---|---|---|---|
| 17268729 | *Fbxl20* | 0.0001 | 7.66 | 72.8 | 0.338 | 0.0287 | 3.35 | 74.6 | 0.024 |

Exp, expression level; Amp, amplitude.

**Source data 1.** Microarray data for MetaCycle analysis.

have been targeted (LNds). We cannot formally exclude the possibility that expression of CLKΔ or CYCΔ in non-clock cells or even in the gut (**Brown et al., 1999**) contributes to this phenotype; however, the effect of targeting CLKΔ to specific LN$_d$s with the *Dvpdf* driver suggests that these cells contribute to the peripheral rhythm phenotype. In addition, even though NPF expression has been reported in both the LN$_d$s and LNvs (**Hermann et al., 2012**), it is unlikely the LNvs regulate *sxe1* rhythms, because disrupting clocks in PDF-positive LNvs does not abolish *sxe1* oscillations. LN$_d$s can be synchronized by inputs from LNvs (**Guo et al., 2014**), but cell-autonomous entrainment mechanisms in the LN$_d$s may limit the influence of LNvs in light:dark cycles, which may explain why ablating clocks in LNvs has a small effect on *sxe1* expression. Thus, we suggest that the clocks in LN$_d$s are required for cycling of *sxe1* and *Cyp6a21* expression in the fat body.

NPF neuropeptide reportedly modulates rest:activity rhythms in *Drosophila* (**Hermann et al., 2012**; **He et al., 2013a**). We did not detect a role for clocks in NPF cells, nor for the single known NPF receptor, in the regulation of rest:activity, but it is possible that other mechanisms are utilized. However, we show that NPF regulates the expression of circadian genes in the fat body. Consistent with the assertion that NPF is the relevant output for fat body rhythms from NPF-positive cells, we also found that flies lacking functional clocks in these cells display significantly reduced *npf* levels (**Figure 4D**). Interestingly, Lee et al. previously showed that *npf* mRNA is absent in the LN$_d$s of adult male *Clk^irk* mutant brains (**Lee et al., 2006**). This further supports our hypothesis that NPF is regulated by the circadian clock in LN$_d$s, and its release from these neurons is necessary for mRNA rhythms of specific fat body genes. However, the effect of NPF on the fat body is likely not direct. Some insect species release NPF into the hemolymph to reach other tissues, but this does not appear to be the case in *Drosophila* (**Nässel and Wegener, 2011**). The NPF receptor may function in clock neurons in the dorsal fly brain (i.e. DN1s), neurons in the suboesophageal ganglion, or

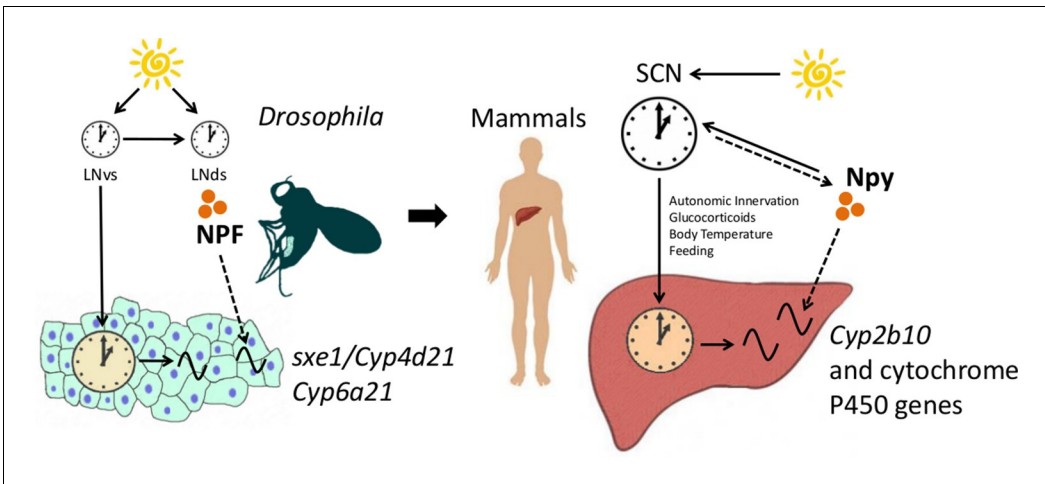

**Figure 6.** NPF/Npy regulate rhythmically expressed P450 enzymes in the periphery of flies and mammals. A model of brain clock regulation of peripheral cycling. Brain clocks regulate clocks in peripheral tissues. In *Drosophila*, clocks in PDF-positive neurons (LNvs) regulate the clock in the fat body. Similarly, in mammals, clocks in the suprachiasmatic nuclei (SCN) have been shown to regulate peripheral clocks such as the liver clock via autonomic innervation, glucocorticoids, body temperature, and feeding. In both the fat body and liver, not all circadian transcripts depend on the local-tissue clock. Clocks in NPF-positive $LN_ds$ and NPF itself regulate circadian expression of cytochrome P450 enzymes in the fly fat body. The LNvs can influence other brain clocks (such as the $LN_ds$), but are not required for rhythms of fat body transcripts in LD as $LN_ds$ may entrain directly to light. In mammals, Npy was previously known to be a non-photic signal involved in entraining the SCN. However, the SCN could also influence Npy production or release, which in turn drives rhythmic expression of cytochrome P450 enzymes in the liver.

neurons innervating the mushroom body (*Krashes et al., 2009*; *He et al., 2013a*). Alternatively, NPF could signal through recently identified neurons downstream of the clock network, which are part of the circadian output circuit driving rest:activity rhythms (*Cavanaugh et al., 2014*). Although much is known about the neuronal clock network, very little is known about the neurons and signals downstream of the clock network, which make up the output pathways leading to rhythms in behavior and physiology. Our discovery that NPF-positive clock neurons drive rhythmic gene expression in the fat body provides a unique opportunity to investigate the pathway(s) that convey circadian information from the brain to peripheral tissues.

We report a striking parallel in the mammalian system, where the NPF ortholog, Npy, drives cyclic expression of specific liver genes, notably several in the cytochrome P450 pathway. Npy is not required for free-running rest:activity rhythms in mice, but it promotes phase shifts in these rhythms in response to non-photic stimuli (*Yannielli and Harrington, 2004*; *Maywood et al., 2002*; *Besing et al., 2012*). Behavioral effects of Npy are likely mediated by its brain expression, but as Npy is also expressed in the periphery, it is possible that a peripheral source contributes to cycling in the liver. Regardless, it is clear that Npy has a profound effect on circadian gene expression in the liver.

Since NPF promotes feeding in *Drosophila* larvae (*Wu et al., 2003*; *Wu et al., 2005*; *Lingo et al., 2007*) and Npy does so in mice, it is possible NPF/Npy drive cycling in the fat body/ liver through the regulation of feeding. Feeding is known to be a potent stimulus for metabolic clocks, with circadian gene expression in peripheral tissues driven by restricted feeding cycles in both flies and mammals (*Xu et al., 2011*; *Gill et al., 2015*; *Vollmers et al., 2009*). However, under conditions of *ad lib* food, feeding rhythms in flies are of low amplitude and likely insufficient to drive robust cycling. Consistent with this, while cyclic expression of *Cyp6a21* can be driven by a restricted feeding paradigm, as can the clock in the fat body, cycling is more robust when this paradigm is conducted in wild type versus clockless animals (*Xu et al., 2011*), indicating that its regulation is not through feeding alone. Finally, time restricted feeding experiments of mice do not support the idea that restricted feeding drives cycling of *Cyp2b10* in clockless mice, even though it is sufficient to

maintain rhythms of many other liver genes (*Vollmers et al., 2009*). Thus, while feeding cannot be discounted as an important factor, which may contribute to the cycling of the genes reported here, these genes are unique in their dependence on Npy. Only a limited subset of liver transcripts previously shown to be independent of the liver clock require Npy for cyclic expression (*Kornmann et al., 2007a*). Similarly, several fly genes, for example *sxe2* and *CG17562*, continue to oscillate when CLKΔ is expressed under *Npf*-GAL4 (data not shown). These results suggest there are additional mechanisms regulating circadian rhythms in the fat body/liver. Why would more than one mechanism exist to couple rhythmic gene expression in a specific peripheral tissue to other clocks? One possibility is that different mechanisms regulate distinct phases of circadian gene expression. Alternatively, different mechanisms may couple gene expression to different cell populations, processes, or behaviors.

The functional importance of the interaction between NPF/Npy and fat body/liver genes in the circadian system is unclear. Cytochrome P450 genes, such as *Cyp6a21, sxe1 and Cyp2b10*, are associated with detoxification (*King-Jones et al., 2006*; *Fujii et al., 2008*), which is likely rhythmic, although not yet reported. Overexpression of NPFR in larvae increases foraging behavior as well as consumption of noxious or bitter compounds (*Wu et al., 2005*). Indeed, NPF/Npy signaling is generally associated with an increase in feeding (*Wu et al., 2003*; *2005*; *Lingo et al., 2007*; *Beck, 2006*), which can lead to ingestion of toxic substances. Thus, coordination of feeding with expression of detoxification enzymes, such as *sxe1, Cyp6a21* and *Cyp2b10,* through NPF/Npy may have evolved to promote survival. Large delays between consumption of noxious substances and their removal could affect an animal's health; thus, the need for coordination between clocks in processing such substances. Conservation of cytochrome P450 regulation from flies to mammals supports the idea that neural control of detoxification in the periphery promotes organismal fitness (*Figure 6*).

In this study we exclusively evaluated males, because the initial studies reporting rhythmic gene expression in the presence and absence of the fat body or liver clock in flies or mammals respectively, were based on males (*Xu et al., 2011*; *Kornmann et al., 2007a*). Interestingly, NPF/Npy and *sxe1/Cyp2b10* expression is sexually dimorphic in *Drosophila* (*Lee et al., 2006*; *Fujii et al., 2008*) and mammals (*Lu et al., 2013*; *Karl et al., 2008*; *Urban et al., 1993*), suggesting there may be some gender specificity to this entire pathway. The functional significance of sex-specific regulation is unclear, but indicates that other mechanisms could contribute to such coordination in females.

This work has implications for chronopharmacology, which is based on the circadian timing of drug metabolism, transport, tolerance, and efficacy. Rhythmic expression of genes involved in drug breakdown and absorption in the liver influences drug efficacy and toxicity (*Dallmann et al., 2014*), and loss of such rhythms can have long-term effects on health and lifespan (*Gachon et al., 2006*). Therefore, expression of these genes may be tightly coordinated to optimize drug metabolism, and speaks to the importance of controlling the timing of drugs that have toxic side effects. The role for Npy reported here suggests it could be a potential target for improving drug efficacy and toxicity. Ultimately, understanding circadian rhythms at a systems level, including interactions between tissues and other physiological systems, will be useful from biological and clinical perspectives.

## Materials and methods

### Fly genetics

Flies were grown on standard cornmeal-molasses medium and maintained at 25°C. The following strains were used: Iso31 (isogenic *w1118* stock; (*Ryder et al., 2004*)), *Pdf*-GAL4 (*Renn et al., 1999*), 911-GAL4 (InSITE Library; (*del Valle Rodríguez et al., 2012*)), *Dvpdf*-GAL4; *pdf*-GAL80 (*Guo et al., 2014*), *Clk^jrk* (*Allada et al., 1998*), and UAS-*npf* RNAi (Vienna Drosophila Resource Center #108772). The following flies were obtained from Bloomington Drosophila Stock Center: *Npf*-GAL4 (#25681), UAS-CLKΔ (#36318), UAS-CYCΔ (#36317), *tub*-GAL80^ts (#7018), and *npfr* mutant (#10747).

### Locomotor activity

The previously described Drosophila Activity Monitoring Systems (Trikinetics, Waltham, MA) were used to monitor rest:activity rhythms under constant conditions. Roughly 1 week old male flies were entrained for at least 3 days to 12 hr light: 12 hr dark cycles (LD) and then transferred to constant darkness for at least 7 days. Data were analyzed using ClockLab software (Actimetrics) and

rhythmicity of individual male flies was determined for days 2–7 of DD as described previously (*Williams et al., 2001*).

## Adult fat body collection

Male flies (roughly 4–7 days old) were entrained to a 12:12 LD cycle at 25°C for at least 3 days before they were harvested. The abdominal fat body was obtained by separating the fly abdomen from the rest of the body and then removing all internal organs, leaving the fat body attached to the cuticle to be collected on dry ice for RNA extraction. For *tub*-Gal80^ts experiments, flies were raised at 18°C. Control flies were kept at 18°C, while the experimental flies were shifted to 30°C, the restrictive temperature for Gal80^ts, for at least 4 days before collection.

## Mice husbandry and liver collection

*Npy* knockout mice were obtained from The Jackson Laboratory (004545) along with their background strain for controls (002448). Genotyping primers are listed on the Jackson website. 8–12 weeks old male mice were entrained to 12:12 LD cycles and fed a standard ad lib diet. Livers from *Npy* knockouts and their background controls were collected every 4 hr starting at lights on (ZT0) and immediately frozen in liquid nitrogen. 3–4 male mouse livers were collected at every timepoint for each genotype. All procedures were approved by the University of Pennsylvania Institutional Animal Care and Use Committee.

## Real-time quantitative PCR and statistical analyses

For each time point, fat bodies from 12 male flies were collected for RNA preparation. Total RNA was extracted using Trizol reagent (Life Technologies, Grand Island, NY) and purified using RNeasy Mini Kit (Qiagen Inc., Valencia, CA) according to manufacturer's protocol. All RNA samples were treated with RNase-free DNase (Qiagen Inc.). RNA was reverse transcribed to generate cDNA using a High Capacity cDNA Reverse Transcription kit (Life Technologies, Grand Island, NY). Quantitative RT-PCR was performed on a 7900HT Fast-Real-Time PCR (Applied Biosystems) using SYBR Green (Life Technologies). The following primer sequences were used for qPCR: $\alpha tubulin$ (Forward 5' CG TCTGGACCACAAGTTCGA 3' and reverse 5' CCTCCATACCCTCACCAACGT 3'), *per* (Forward 5' CGTCAATCCATGGTCCCG 3' and reverse 5' CCTGAAAGACGCGATGGTG 3'), *Cyp4d21/sxe1* (Forward 5' CTCCTTTGGTTTATCGCCGTT 3' and reverse 5' TTATCAGCGGCTTGTAGGTGC), *sxe2* (Forward 5' TGCGGTACGATCTTTATACGCC 3' and reverse 5' CTAACTGGCCATTTCGGATTGA 3'), *CG14934* (Forward 5' GGAAATCACGACAATCCTCGA 3' and reverse 5' CCCAACTCCTCGCCATTA TAAG 3'), *Cyp6a21* (Forward 5' GTTGTATCGGAAACCCTTCGATT 3' and reverse 5' AACCTCATAG TCCTCCAGGCATT 3'), and *CG117562* (Forward 5'ACCACAGAGGTGAAACGCATCT 3' and reverse 5'CAGCAGCAGTTCAAATACCGC 3'). Transcript levels were normalized to those of $\alpha tubulin$ to control for the total RNA content in each sample.

Kits and procedures to isolate RNA and generate cDNA from mouse livers are the same as described above for fly fat bodies. The following primer sequences were used for qPCR: *Cyp2b10* (Forward 5' GACTTTGGGATGGGAAAGAG 3' and reverse 5' CCAAACACAATGGAGCAGAT 3'), *36B4* (Forward 5' TCCAGGCTTTGGGCATCA 3' and reverse 5' CTTTATCAGCTGCACATCAC TCAGA 3'), *Rev-erb alpha* (Forward 5' GTCTCTCCGTTGGCATGTCT 3' and reverse 5' CCAAGTTCA TGGCGCTCT 3') and *Alas1* (PrimerBank ID 23956102a1) (*Spandidos et al., 2008*; *2010*; *Wang and Seed, 2003*). Transcript levels were normalized to the housekeeping gene, *36B4*.

Significant circadian rhythmicity of transcript levels was determined using the JTK_Cycle algorithm (*Hughes et al., 2010*). P values of less than 0.05 were considered significant. We also used two-way ANOVA for repeated measures and a Tukey's *post hoc* test for differences across time (GraphPad Prism). P-values are reported in *Table 3*.

## Microarray analysis

Liver samples from *Npy* KO and wild type mice were collected every 4h over 24h (n = 2 per genotype and timepoint). RNA was purified as described above. Expression profiling was done at the Penn Molecular Profiling Facility using Mouse Gene 2.0 ST Arrays (Affymetrix, Santa Clara, CA, which also provided the annotation files). For extracting expression values of transcripts, raw CEL files were analyzed with the RMA algorithm (*Irizarry, 2003*) implemented in the affy package in

**Table 3.** JTK_Cycle Statistics and Two-factor ANOVA Results. All qPCR data were tested for circadian rhythmicity with JTK_cycle test and two-way ANOVA for repeated measures and a Tukey's *post hoc* test. P-values from these tests are summarized.

| Figure | Genotype | Gene | Tissue | JTK_cycle P-value | Time P-value | Genotype P-value | Time X Genotype P-value |
|---|---|---|---|---|---|---|---|
| 1C | [ZT] UAS-CLKΔ/+ | *per* | Fat Body (FB) | 0.0194 | < 0.0001 | 0.7826 | 0.4679 |
| 1C | [ZT] *Pdf*-GAL4/UAS-CLKΔ | *per* | FB | 0.0094 | | | |
| 1D | [CT] UAS-CLKΔ/+ | *per* | FB | 0.0041 | 0.0313 | 0.6650 | 0.0281 |
| 1D | [CT] *Pdf*-GAL4/UAS-CLKΔ | *per* | FB | 1 | | | |
| | | | | | | | |
| 2A | *Iso31* | sxe2 | FB | 0.0014 | 0.0131 | 0.1852 | 0.0843 |
| 2A | *Clk^irk* | sxe2 | FB | 1 | | | |
| 2B | *Iso31* | CG17562 | FB | 0.0014 | 0.0223 | 0.6409 | 0.1746 |
| 2B | *Clk^irk* | CG17562 | FB | 0.6945 | | | |
| 2C | *Iso31* | sxe1 | FB | 0.0020 | 0.0019 | <0.0001 | 0.0020 |
| 2C | *Clk^irk* | sxe1 | FB | 1 | | | |
| 2D | *Iso31* | CG14934 | FB | 0.0820 | 0.1146 | <0.0001 | 0.0266 |
| 2D | *Clk^irk* | CG14934 | FB | 0.5429 | | | |
| | | | | | | | |
| 3A | UAS-CLKΔ/+ | sxe1 | FB | 0.0007 | 0.0003 | 0.0049 | 0.7061 |
| 3A | *Pdf*-GAL4/UAS-CLKΔ | sxe1 | FB | 9.00E-07 | | | |
| 3B | UAS-CYCΔ/+ | sxe1 | FB | 0.0363 | 0.0006 | 0.7995 | 0.9592 |
| 3B | 911-GAL4/UAS-CYCΔ | sxe1 | FB | 0.0044 | | | |
| 3C | UAS-CLKΔ/+ | sxe1 | FB | 0.0014 | 0.0060 | <0.0001 | 0.0749 |
| 3C | *Npf*-GAL4/UAS-CLKΔ | sxe1 | FB | 0.1969 | | | |
| 3D | UAS-CYCΔ/+ | sxe1 | FB | 0.0029 | <0.0001 | <0.0001 | <0.0001 |
| 3D | *Npf*-GAL4/UAS-CYCΔ | sxe1 | FB | 1 | | | |
| 3E | UAS-CLKΔ/+ | sxe1 | FB | 0.0196 | 0.0038 | 0.0017 | 0.5326 |
| 3E | *Dvpdf*-GAL4/UAS-CLKΔ;*Pdf*-GAL80/+ | sxe1 | FB | 0.0001 | | | |
| 3F | UAS-CLKΔ/+ | Cyp6a21 | FB | 0.0009 | <0.0001 | <0.0001 | 0.0574 |
| 3F | *Npf*-GAL4/UAS-CLKΔ | Cyp6a21 | FB | 0.0259 | | | |
| 3G | UAS-CLKΔ/+ | *per* | FB | 0.0568 | 0.0057 | 0.8767 | 0.9902 |
| 3G | *Npf*-GAL4/UAS-CLKΔ | *per* | FB | 0.0441 | | | |
| | | | | | | | |
| 4A | UAS-Npf RNAi/+ | sxe1 | FB | 0.0020 | 0.0005 | <0.0001 | 0.1421 |
| 4A | *Npf*-GAL4/UAS-Npf RNAi | sxe1 | FB | 0.0441 | | | |
| 4B | Not analyzed with JTK | npf | Head | - | | | |
| 4C | *npfr*/+ | sxe1 | FB | 0.0128 | 0.0008 | <0.0001 | 0.0038 |
| 4C | *npfr* | sxe1 | FB | 0.1969 | | | |
| 4D | *npfr*/+ | Cyp6a21 | FB | 0.0916 | 0.0319 | <0.0001 | 0.4182 |
| 4D | *npfr* | Cyp6a21 | FB | 0.1150 | | | |
| 4E | UAS-CLKΔ/+ | npf | Head | 1 | 0.7588 | <0.0001 | 0.846 |
| 4E | *Npf*-GAL4/UAS-CLKΔ | npf | Head | 1 | | | |
| 4F | UAS-CLKΔ/+ | *per* | Head | 0.0001 | <0.0001 | 0.1410 | 0.0420 |
| 4F | *Npf*-GAL4/UAS-CLKΔ | *per* | Head | 1.19E-05 | | | |
| | | | | | | | |
| 5A | Wild type | Cyp2b10 | Liver | 0.0010 | 0.0209 | 0.0034 | 0.1640 |

*Table 3 continued on next page*

*Table 3 continued*

| Figure | Genotype | Gene | Tissue | JTK_cycle P-value | Time P-value | Genotype P-value | Time X Genotype P-value |
|--------|----------|------|--------|-------------------|--------------|------------------|-------------------------|
| 5A | *Npy* Knockout (KO) | *Cyp2b10* | Liver | 0.0510 | | | |
| 5B | Wild type | *Reverbα* | Liver | 7.46E-12 | <0.0001 | 0.4979 | 0.0661 |
| 5B | *Npy* KO | *Reverbα* | Liver | 7.17E-09 | | | |
| 5C | Wild type | *Alas1* | Liver | 9.81E-06 | 0.0007 | 0.8476 | 0.8202 |
| 5C | *Npy* KO | *Alas1* | Liver | 0.0018 | | | |

Bioconductor in R (R 2.14.2) (*Gautier et al., 2004*). The newly developed MetaCycle (version 1.0.0; https://github.com/gangwug/MetaCycleV100.git) was used to detect circadian transcripts from time-series expression data in the wild type (WT) and *Npy* knockout (KO) groups, respectively. Key parameters in MetaCycle were the periodicity detection algorithms, JTK_CYCLE (*Hughes et al., 2010*) and Lomb-Scargle (*Glynn et al., 2006*), the period length (set at exactly 24 hr), and the p-value integration method (Fisher's method, Fisher 1956). Using MetaCycle, we calculated two new features of circadian transcripts, baseline expression level (bEXP) and relative amplitude (rAMP). The former one is defined as the average expression level of a cycling transcript within one period length, and the latter one is a normalized amplitude value with bEXP. Based on analysis results from MetaCycle, expressed transcripts (bEXP larger than 101.6) with a p-value <0.01 in WT and >0.8 in the KO group were considered WT-specific rhythmic transcripts and shown in the heatmap. To generate the heatmap, expression values from replicate libraries in each group were averaged, median normalized by transcript, sorted by phase, and plotted as a heatmap using pheatmap in R.

## Acknowledgements

We thank members of the Sehgal laboratory for input and advice throughout this project. We are also grateful for the services provided by the Neurobehavioral Testing core at the UPenn Smilow Center. The work was supported by NIH grant R37NS048471. RE and ANK were supported in part by a Predoctoral Training Grant in Genetics (T32 GM008216).

## Additional information

### Funding

| Funder | Grant reference number | Author |
|--------|------------------------|--------|
| National Institutes of Health | NS048471 | Amita Sehgal |
| National Institute of General Medical Sciences | T32 GM008216 | Renske Erion Anna N King |

The funders had no role in study design, data collection and interpretation, or the decision to submit the work for publication.

### Author contributions

RE, Conception and design, Acquisition of data, Analysis and interpretation of data, Drafting or revising the article; ANK, Acquisition of data, Analysis and interpretation of data, Drafting or revising the article; GW, JBH, Analysis and interpretation of data, Drafting or revising the article; AS, Conception and design, Analysis and interpretation of data, Drafting or revising the article

### Author ORCIDs

Amita Sehgal, http://orcid.org/0000-0001-7149-8588

### Ethics

Animal experimentation: This study was performed in strict accordance with the recommendations in the Guide for the Care and Use of Laboratory Animals of the National Institutes of Health. All of the animals were handled according to approved institutional animal care and use committee (IACUC) protocols of the University of Pennsylvania.

## Additional files

### Supplementary files

• Supplementary file 1. MetaCycle statistics of microarray data generated in this study. MetaCycle p-value, phase, baseline expression (baselineEXP), and relative amplitude (relativeAMP) are calculated for each gene in wild type and *NPY* KO animals.

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
