## [Decision Letter]

Thank you for submitting your article "Neural clocks and NPF/Npy regulate circadian gene expression in a peripheral metabolic tissue" *eLife*for consideration by eLife. Your article has been reviewed by two peer reviewers, and the evaluation has been overseen by K VijayRaghavan as the Senior and Reviewing Editor.

The reviewers have discussed the reviews with one another and the Reviewing Editor has drafted this decision to help you prepare a revised submission.

Summary:

In this manuscript, Erion et al. study an important issue in circadian biology: how peripheral circadian oscillators communicate with the main pacemaker in *Drosophila*. The main claim in the manuscript is that *sxe1* and *Cyp6a21*, two genes that cycle independently of the fat body clock require clocks in other tissues. The authors report that both genes are regulated by NPF-expressing neurons which are themselves regulated by CLK. Moreover, the authors present data showing that the mammalian homolog of NPF (Npy) regulates P450 genes in the mouse liver, suggesting that this mechanism is evolutionary conserved. Although the manuscript provides new and interesting data, we have a concerns regarding the interpretation and significance of the data. These are given below. If these important concerns are addressed, this will be a strong addition to the literature. The inclusion of fly and mouse data makes a strong statement and the interesting relationship between circadian timing, metabolism and behaviour becomes more intriguing as it is detailed with genes like *npy* and *sxe1*.

Essential revisions:

Figure 3 shows that adult ablation of NPF-positive neurons is not sufficient to affect *sxe1* rhythms. Unless we are missing something, does this not contradict the main model stating a relationship between NPF and *sxe* in adults? A clear explanation for this should be provided.

Another important issue is whether the effect of the brain clock over *sxe1* and *Cyp6a21* mRNA level and oscillations are direct through NPF or mediated by another oscillating behavior such as feeding or mating. Determining whether time-restricted feeding (TRF) can have an effect in the phase and amplitude of *sxe1* and *Cyp6a21* rhythms in the NPF-DClock flies will help determine whether the observed effects could be due to secondary effects related to feeding or not. Even understanding how TRF affect *sxe1* and *Cyp6a21* oscillations could be illuminating in this aspect and provide support for the main hypothesis on the manuscript. Further, it is indeed surprising that the authors don't use the data in Gill et al. (Science, 2014) to determine if this is the case.

The authors clearly demonstrate that the oscillations of *sxe1* and *Cyp6a21* are abolished on *Clk^Jrk^* mutants and that *Npf* itself it is regulated by CLK. However, these flies have multiple defects and determining the same parameters in *tim* or *per* mutants would be more informative. The authors should either speedily include these results or explain why they are not necessary.

[Editors’ note: a previous version of this study was rejected after peer review, but the authors submitted for reconsideration. The previous decision letter after peer review is shown below.]

Thank you for choosing to send your work entitled "Brain clocks and NPF regulate circadian gene expression in a peripheral metabolic tissue" for consideration at *eLife*. Your full submission has been evaluated by a Senior editor and three peer reviewers, one of whom was also guest Reviewing editor, and the decision was reached after discussions between the reviewers. Based on our discussions and the individual reviews below, we regret to inform you that your work will not be considered further for publication in *eLife*.

This is an interesting paper from Sehgal and colleagues, which concerns the control of circadian cycling of fat body gene expression by the brain clock in *Drosophila*. Previously, the Sehgal lab has shown that 40% of cycling fat body transcripts oscillate independently from the local fat body clock, i.e., they are driven by factors external to the fat body clock. The situation is vaguely similar to mammalian liver. The claim now is that the neuropeptide NPF is produced from some clock neurons and regulates the fat body clock. However, all three reviewers had only muted enthusiasm for the current version of this paper. They were also unanimous in wanting a lot more data to support the claim and to raise the profile of the paper. It seemed unlikely to all the reviewers that the authors would be able to gather these data in the requisite 2-month time interval recommended by *eLife*. Moreover, the authors run a substantial risk that the data will change major conclusions of the paper. So with these issues in mind, the reviewers agreed that a rejection decision is more appropriate than revise and that the authors return to *eLife* with a new manuscript if that is the route they choose; this will obviate any temporal issues.

*Reviewer #1:*

This paper from Sehgal and colleagues concerns the control of the fat body clock by the brain clock in *Drosophila*. The story is that the neuropeptide NPF is produced from some clock neurons, and it regulates the fat body clock. However, the focus is on a single fat body transcript (out of the 40% that have been reported to oscillate independently from the fat body clock), and it is unclear that this result is applicable to a wider set of mRNAs. The data also make me concerned about developmental effects. Moreover, I am not convinced that any aspect of the fat body clock is influenced by NPF – as distinct from some other effect on gene expression of a rather limited subset of fat body mRNAs. The authors also admit that the effect of NPF on the fat body is almost certainly indirect (through feeding?), so we do not know anything mechanistic about how NPF is exercising its influence.

I hope the specific comments will be of help to the authors.

1) In the subsection “The Central Clock Regulates the Fat Body Clock in Constant Darkness”: It would have been much better to use another tool rather than CLK∆, a tool that isn't so disruptive of gene expression. Also, the authors should definitely use gene switch (GS) to avoid developmental effects.

2) Figure 1: There are poor fat body rhythms already at DD2 in WT. So they are indeed worse without PDF neuron function, but this is on the whole a quite unimpressive result. The effect is therefore too weak to make such a strong claim, especially without a GS driver and an immediate assay after feeding drug.

3) The three scenarios the authors describe in the subsection “Rhythmic Expression of Fat Body Transcripts that Cycle Independently of the Local Tissue Clock Require Organismal Circadian Function” include a straw man. The effect of nutrients on the fat body clock depends on the robustness of feeding rhythms, which are rather weak in flies.

4) The *Clk^jrk^* effect could be developmental. This could be addressed (imperfectly but nonetheless appropriately) with *elav* GS.

5) The authors describe the level of 2 of the 4 fat body cycling genes (externally driven) as "attenuated." This is inaccurate in my view, as mRNA levels are virtually absent. And it is quite troubling that *sxe1*, the gene that they focus on for the rest of the paper just isn't expressed in *Clk^jrk^*. So how does this relate to circadian oscillations?

6) The authors spend a large amount of text describing negative results, the different clock neuron drivers that have no effect on rhythmicity. It is strange that they do not also use the more typical Clk 4.1 DN1 driver.

7) The positive result is with the *Npf-GAL4* driver. However, disrupting the clock in these neurons doesn't really eliminate rhythmic expression of *sxe1* mRNA as much as it dramatically lowers expression, i.e., I can imagine a rhythm with very low levels of mRNA.

8) There are several other difficulties with this result. First there is the usual developmental effects possibility, i.e., no use of gene switch or *tub*-GAL80^ts^ to reduce the influence of these effects. Second there is no attempt to address clock neuron specificity. (Could there be relevant CLK and CYC expression in non-clock neurons? Would the addition of cry-GAL80 be informative?)

9) The data in Figure 4 further suggest a developmental effect. This would explain why expression of CLK∆ and the NPFR strain have strong effects (on *sxe1* expression levels, not necessarily rhythmicity) whereas the knockdown is much less potent.

10) Unfortunately, the mammalian result suffers from similar criticisms. There are lower levels of *cyp2b10* mRNA but still some cycling to my eye. Moreover, this is a systemic KO and all through development.

*Reviewer #2:*

The manuscript by Erion et al., addresses an important issue in circadian biology: the communication between central and peripheral circadian oscillators in *Drosophila*. While I find the work solid and important for the field, I have a few concerns about the interpretation and significance of the data, especially in light of a recent manuscript from the Panda lab (Gill et al., Science 2014). The main claim in the manuscript is that *sxe1* and *sxe2*, two genes that cycle independently of the fat body clock require clocks in other tissues. This seems more prominent for *sxe1*, which they report is regulated by NPF-expressing neurons.

The main issue is whether the effect of the brain clock over *sxe1* mRNA level and oscillations is direct through NPF or mediated by another oscillating behavior like feeding or mating. I think that determining whether time-restricted feeding (TRF) can have an effect in the phase and amplitude of *sxe1* rhythms in the NPF-DClock flies will help determine whether the observed effects could be due to secondary effects related to feeding or not. Even understanding how TRF affect *sxe1* oscillations could be illuminating in this aspect. I looked into the Gill et al. manuscript, but unhappily *sxe1* was not found among the oscillating mRNAs neither in heads or periphery. This is likely due to difference in strains.

In the same line as above, some of the Npf-related effects might be developmental and experiments using induced expression of DClk in the NPF cells (i.e. by the use of the GAL80ts system) could rule out this possibility.

Despite my concern stated above, the authors find good support for their hypothesis that NPF neurons regulate *sxe1* oscillations. However, I would be surprised if this will be the only gene regulated in this way and the manuscript will gain a lot from the description of a second gene in this route.

Based on the experiment performed in Figure 4 the authors suggest that Clk regulates NPF levels. However, the effect could be due also to effects of CLK∆ in gene expression in cells that normally don't express Clk. In order to rule this out the authors should show that NPF mRNA levels are also downregulated in Clk mutants.

*Reviewer #3:*

This study is a follow-up to the author's earlier manuscript showing that ~40% of oscillating transcripts in the *Drosophila* fat body continue to oscillate in the absence of a functional circadian oscillator in the fat-body. Although food, temperature cycle, and light can drive oscillations in the absence of a local peripheral clock, the authors focused on specific transcripts that cannot be driven by these stimuli and hence were dependent on clocks elsewhere. They have systematically addressed this issue by examining the fat-body oscillation of *sxe* genes in fly strains lacking circadian clock in specific neurons and have found the NPF expressing neurons and the NPF signaling system somehow drives peripheral oscillations. To test the relevance of this regulation in mammals, they have tested *Cyp10b2* gene expression in the liver of NPY knockout mice and have shown dampened oscillation. The authors imply that this is the first time a brain derived neuropeptide drives circadian oscillation in a peripheral organ and suggest NPY may be targeted to improve circadian regulation of xenobiotic metabolism.

Most of the experiments are straightforward in which the authors test mRNA expression of candidate gene by qPCR in various fly lines or liver gene expression in NPY knockout mice.

Given the role of NPY and NPF in feeding, and feeding is known to drive ~24 hr oscillation in gene expression in the complete absence of a clock anywhere in the body, it is important to have some measure of feeding pattern in these experiments in both flies and mice.

In rodents, NPY is also expressed in peripheral tissues including the gut. A recent paper from Chang's lab has shown the gut plays a profound role in driving liver circadian gene expression. Therefore, it is premature to state the CNS expression of NPY is driving liver circadian gene expression. The authors may use a hypothalamus specific knockout of NPY if they want to drive this message. Along the same line, it will be worth examining whether NPF system is expressed in the peripheral organs of flies.

With the lab's track record in genome wide studies and this paper being a follow-up of their earlier gene expression studies in fat body, it is worth including a RNA-seq or microarray study to examine to what extent the NPF system in flies and NPY system in mice regulate circadian fatbody/liver gene expression independent of feeding pattern. Such experiments will expand the impact and relevance of the paper beyond just a couple of liver transcripts.

---

## [Author Response]

Essential revisions:

Figure 3 shows that adult ablation of NPF-positive neurons is not sufficient to affect sxe1 rhythms. Unless we are missing something, does this not contradict the main model stating a relationship between NPF and sxe in adults. A clear explanation for this should be provided. The reviewer is correct in that adult ablation of NPF cells does not affect *sxe1* rhythms. However, it does affect cycling of the other cytochrome P450 gene we studied, *cyp6a21* (Figure 3). These findings indicate that effects of NPF on circadian rhythms range from developmental to adult function. Our main model did not invoke an adult-specific role for NPF, but we can further clarify this in the revised manuscript.

Another important issue is whether the effect of the brain clock over sxe1 and Cyp6a21 mRNA level and oscillations are direct through NPF or mediated by another oscillating behavior such as feeding or mating. Determining whether time-restricted feeding (TRF) can have an effect in the phase and amplitude of sxe1 and Cyp6a21 rhythms in the NPF-DClock flies will help determine whether the observed effects could be due to secondary effects related to feeding or not. Even understanding how TRF affect sxe1 and Cyp6a21 oscillations could be illuminating in this aspect and provide support for the main hypothesis on the manuscript. Further, it is indeed surprising that the authors don't use the data in Gill et al. (Science, 2014) to determine if this is the case.

We include a time-restricted feeding (TRF) experiment (Figure 7), and list the reasons we believe that although feeding can affect cycling of *sxe1*, it is not the only, and most likely not even the major, contributor to rhythms of genes driven by NPF/NPY:

Author response image 1.Restricted feeding between ZT9 and ZT15 alters the rhythmic expression of *sxe1* and clock gene *per* in the *Drosophila* fat body.**DOI:**
http://dx.doi.org/10.7554/eLife.13552.021

We conducted time-restricted feeding (TRF) experiments and assayed effects on cycling of *sxe1* as well as of the *period (per)* clock gene. We found that restricting food to a time of day when flies do not normally feed alters the phase of both *per* and *sxe1* cycling (Figure 7). Thus, *sxe1* is sensitive to feeding, but not more so than *per*, which is not regulated by NPF. More likely, the shift in the fat body clock (*per*) shifts *sxe1* expression even though *sxe1* cycling does not require the fat body clock.

We have found that effects of NPF on *sxe1* are developmental (as noted above), so NPF could not be driving rhythms of *sxe1* through cycles of feeding.

Regarding *Cyp6a21*, we showed previously that while it can be driven by restricted feeding (RF) in *Clk^Jrk^* flies the rhythm is not as robust as seen upon RF in wild type flies (Xu et al., 2011). Thus, it does not appear to be driven by a feeding cycle alone. This is consistent with the fact, also pointed out by previous Reviewer 1, that feeding cycles are quite low in amplitude in flies and unlikely to account for robust molecular rhythms.

We checked the data of Gill et al. as suggested by the reviewer. They show that *sxe1* cycling is not regulated by feeding.

The Panda database (http://circadian.salk.edu/about.html) does not indicate restoration of *Cyp2b10* cycling by restricted feeding of clockless mice.

We note too that only a limited subset of circadian genes that are independent of the liver clock depend on NPY. If NPY were driving gene expression through rhythmic feeding alone, it would impact most liver genes.

The authors clearly demonstrate that the oscillations of sxe1 and Cyp6a21 are abolished on Clk^Jrk^ mutants and that Npf itself it is regulated by CLK. However, these flies have multiple defects and determining the same parameters in tim or per mutants would be more informative. The authors should either speedily include these results or explain why they are not necessary.

We refer the reviewer to an earlier paper from the Young laboratory (Claridge-Chang et al., 2001), which showed that *sxe1 (cyp4d21)* and *cyp6a21* mRNAs cycle in wild type but not in three different clock mutants – *per^0^, tim^01^*and *Clk^jrk^*. Claridge-Chang et al. did not detect cycling of *Npf* mRNA and neither did we (Figure 4). While NPFis probably rhythmic on some level –perhaps in some specific cells, or in terms of protein expression or release –the RNA is likely not under clock control globally. The link below indicates that it is significantly affected in *tim^01^* mutants, but not in *per^0^* and *Clk^Jrk^.*

See the webpages on each gene for data and graphs:

sxe1 (cyp4d21): http://biorhythm.rockefeller.edu/probeset.php?probeset=152958_at

cyp6a21: http://biorhythm.rockefeller.edu/probeset.php?probeset=142190_at

npf: http://biorhythm.rockefeller.edu/probeset.php?probeset=144175_at

[Editors’ note: the author responses to the previous round of peer review follow.]

*Reviewer #1: This paper from Sehgal and colleagues concerns the control of the fat body clock by the brain clock in Drosophila. The story is that the neuropeptide NPF is produced from some clock neurons, and it regulates the fat body clock. However, the focus is on a single fat body transcript (out of the 40% that have been reported to oscillate independently from the fat body clock), and it is unclear that this result is applicable to a wider set of mRNAs. The data also make me concerned about developmental effects. Moreover, I am not convinced that any aspect of the fat body clock is influenced by NPF – as distinct from some other effect on gene expression of a rather limited subset of fat body mRNAs. The authors also admit that the effect of NPF on the fat body is almost certainly indirect (through feeding?), so we do not know anything mechanistic about how NPF is exercising its influence.* We agree with the reviewer that NPF does not influence the *clock* in the fat body, but rather a “limited subset of fat body mRNAs”. Indeed, we make the case in the manuscript that the clock is *not*affected, as *per* cycling is normal when the clock is disrupted in NPF cells. ~30 transcripts in the fat body cycle independently of the fat body clock, and appear to be controlled by different mechanisms. For instance, in other work we have found that cycling at least one of these transcripts requires insulin-producing cells. The transcripts reported in this manuscript, *sxe1* and a second characterized in the revised manuscript, *cyp6a21*, depend on NPF. As noted below, we have now conducted experiments to address the developmental issue raised by the reviewer. The identification of NPF as a neural factor that drives rhythmic expression in a metabolic tissue provides a mechanism for a longstanding question of interest in the circadian field, namely how coordinated action of clocks in different tissues regulates circadian outputs (metabolic gene expression in this case).

Exactly how NPF does so is, we believe, beyond the scope of this manuscript. As discussed in detail in response to Reviewer 3 (and also in the manuscript in the sixth paragraph of the Discussion), feeding may contribute to the effects of NPF/NPY, but it is unlikely to be the only mechanism.

*1) In the subsection “The Central Clock Regulates the Fat Body Clock in Constant Darkness“: It would have been much better to use another tool rather than CLK*∆*, a tool that isn't so disruptive of gene expression. Also, the authors should definitely use gene switch (GS) to avoid developmental effects.*

We also shown data with dominant negative CYC. Unfortunately, no other tool effectively disrupts the clock. In response to the reviewer’s question, we conducted experiments to avoid developmental effects. As a gene switch (GS) driver was not available for NPF cells and we felt that disrupting the clock in all neurons (with *elav* GS) constituted too blunt a tool, we used the longer and more laborious method of coupling *Npf-*GAL4 with temperature sensitive tubulin-GAL80. Restricting clock disruption in NPF cells to the adult stage did not affect *sxe1* cycling, suggesting that the effect is developmental. However, it did affect cycling of *cyp6a21*. We conclude that NPF has differential effects on rhythmically expressed transcripts in the fat body.

2) Figure 1: There are poor fat body rhythms already at DD2 in WT. So they are indeed worse without PDF neuron function, but this is on the whole a quite unimpressive result. The effect is therefore too weak to make such a strong claim, especially without a GS driver and an immediate assay after feeding drug.

We agree and, therefore, have toned down the conclusions of this experiment.

3) The three scenarios the authors describe in the subsection “Rhythmic Expression of Fat Body Transcripts that Cycle Independently of the Local Tissue Clock Require Organismal Circadian Function “include a straw man. The effect of nutrients on the fat body clock depends on the robustness of feeding rhythms, which are rather weak in flies.

True. We have eliminated specific consideration of this scenario, which, in any case, turned out to be irrelevant.

4) The Clk^jrk^ effect could be developmental. This could be addressed (imperfectly but nonetheless appropriately) with elav GS.

As noted above, we have addressed the developmental issue with a better tool than *elav* GS.

5) The authors describe the level of 2 of the 4 fat body cycling genes (externally driven) as "attenuated." This is inaccurate in my view, as mRNA levels are virtually absent. And it is quite troubling that sxe1, the gene that they focus on for the rest of the paper just isn't expressed in Clk^jrk^. So how does this relate to circadian oscillations?

We have changed “attenuated” to “greatly reduced”. As noted, we have added analysis of a second transcript, *cyp6a21*, which is expressed at reasonable levels when the clock in NPF cells is disrupted, but fails to cycle. Together with the new mouse data, which show effects of NPY loss on the circadian expression of many transcripts, without a significant change in levels of these transcripts, we believe we now present a strong case for effects of NPF/NPY on circadian oscillations.

6) The authors spend a large amount of text describing negative results, the different clock neuron drivers that have no effect on rhythmicity. It is strange that they do not also use the more typical Clk 4.1 DN1 driver.

We did try the Clk 4.1 driver, but failed to get consistent results. In our experience, the Clk 4.1 DN1 is a weak driver and so we switched to the stronger 911 driver. However, as noted in the text, even this driver did not indicate a role for DN1s.

7) The positive result is with the NPF-gal4 driver. However, disrupting the clock in these neurons doesn't really eliminate rhythmic expression of sxe1 mRNA as much as it dramatically lowers expression, i.e., I can imagine a rhythm with very low levels of mRNA.

As noted above, manipulations of NPF cells do greatly reduce levels of *sxe1* so an effect on levels alone cannot be excluded (note that *Clk^jrk^*mutants show similar levels of *sxe1* so this most likely reflects the effect of loss of the clock on *sxe1*). However, the revised manuscript shows that disrupting the clock in NPF cells clearly dampens cycling of *cyp6a21.*

8) There are several other difficulties with this result. First there is the usual developmental effects possibility, i.e., no use of gene switch or tub-GAL80^ts^ to reduce the influence of these effects. Second there is no attempt to address clock neuron specificity. (Could there be relevant CLK and CYC expression in non-clock neurons? Would the addition of cry-GAL80 be informative?)

We have addressed a possible developmental effect through the use of *tub*-GAL80 (see above). Also, the manuscript demonstrates a role for the LN_d_ clock neurons through the useof the *DvPdf* driver.

9) The data in Figure 4 further suggest a developmental effect. This would explain why expression of CLK∆ and the NPFR strain have strong effects (on sxe1 expression levels, not necessarily rhythmicity) whereas the knockdown is much less potent.

As our new data indicate, the effect of NPF on *sxe1* is indeed developmental. However, adult-specific manipulation of NPF cells clearly has an effect on cycling of *cyp6a21.*

*10) Unfortunately, the mammalian result suffers from similar criticisms. There are lower levels of cyp2b10 mRNA but still some cycling to my eye. Moreover, this is a systemic KO and all through development.* In response to Reviewer 3’s last comment, we conducted microarray analysis to compare cycling gene expression in wild type and *NPY* knockout livers, and find that many liver transcripts require NPY for rhythmicity. We show also that average levels of transcripts are not altered in NPY livers.

Reviewer #2:

The manuscript by Erion et al., addresses an important issue in circadian biology: the communication between central and peripheral circadian oscillators in Drosophila. While I find the work solid and important for the field, I have a few concerns about the interpretation and significance of the data, especially in light of a recent manuscript from the Panda lab (Gill et al., Science 2014). The main claim in the manuscript is that sxe1 and sxe2, two genes that cycle independently of the fat body clock require clocks in other tissues. This seems more prominent for sxe1, which they report is regulated by NPF-expressing neurons.

The main issue is whether the effect of the brain clock over sxe1 mRNA level and oscillations is direct through NPF or mediated by another oscillating behavior like feeding or mating. I think that determining whether time-restricted feeding (TRF) can have an effect in the phase and amplitude of sxe1 rhythms in the NPF-DClock flies will help determine whether the observed effects could be due to secondary effects related to feeding or not. Even understanding how TRF affect sxe1 oscillations could be illuminating in this aspect. I looked into the Gill et al. manuscript, but unhappily sxe1 was not found among the oscillating mRNAs neither in heads or periphery. This is likely due to difference in strains.

As suggested by the reviewer, we conducted time-restricted feeding (TRF) experiments and assayed effects on cycling of *sxe1* as well as of the *period (per)* clock gene. We found that restricting food to a time of day when flies do not normally feed alters the phase of both *per* and *sxe1* cycling (Figure 7). Thus, *sxe1* is sensitive to feeding, but not more so than *per*, which is not regulated by NPF. In addition, as noted below, we have found that effects of NPF on *sxe1* are developmental, so it does not appear that cycles of feeding directly drive cycling of *sxe1*. Nevertheless, we cannot exclude the possibility that at least some effects of NPF on peripheral rhythms are mediated through feeding. (See also the response to Reviewer 3’s first point)

(A) The restricted feeding procedure for flies. Food was restricted to ZT9-ZT15, the trough of feeding determined in Xu et al. 2008. Flies were transferred to a 1% agar vial from a normal food vial on day 1. Restricted feeding was started on day 2 and continued until day 7 at which point the flies were collected. (B, C) Quantitative PCR was performed to assay expression of *sxe1* (B)and *per* (C)at different times of day in the fat body of flies maintained on ad lib food (blue) or restricted to food between ZT9 and ZT15 (red). Expression values are normalized to α-tubulin. Each data point represents the average of *n*=2-3 independent samples, and the error bars represent SEM.

In the same line as above, some of the Npf-related effects might be developmental and experiments using induced expression of DClk in the NPF cells (i.e. by the use of the GAL80ts system) could rule out this possibility.

Upon the reviewer’s recommendation, we used the GAL80ts system to restrict expression of DClk to adult NPF cells. We show in the revised manuscript that effects of NPF on *sxe1* are developmental. However, a new gene we have characterized in the revised manuscript, *cyp6a21*, is regulated by NPF in adults.

Despite my concern stated above, the authors find good support for their hypothesis that NPF neurons regulate sxe1 oscillations. However, I would be surprised if this will be the only gene regulated in this way and the manuscript will gain a lot from the description of a second gene in this route.

As suggested, we have included analysis of a second gene, *cyp6a21*.

Based on the experiment performed in Figure 4 the authors suggest that Clk regulates NPF levels. However, the effect could be due also to effects of ClkDelta in gene expression in cells that normally don't express Clk. In order to rule this out the authors should show that NPF mRNA levels are also downregulated in Clk mutants. This is an important point. In fact, it was previously shown that NPF mRNA levels are reduced in *Clk* mutants (Lee, Bahn, and Park 2006).

Reviewer #3:

This study is a follow-up to the author's earlier manuscript showing that ~40% of oscillating transcripts in the Drosophila fat body continue to oscillate in the absence of a functional circadian oscillator in the fat-body. Although food, temperature cycle, and light can drive oscillations in the absence of a local peripheral clock, the authors focused on specific transcripts that cannot be driven by these stimuli and hence were dependent on clocks elsewhere. They have systematically addressed this issue by examining the fat-body oscillation of sxe genes in fly strains lacking circadian clock in specific neurons and have found the NPF expressing neurons and the NPF signaling system somehow drives peripheral oscillations. To test the relevance of this regulation in mammals, they have tested Cyp10b2 gene expression in the liver of NPY knockout mice and have shown dampened oscillation. The authors imply that this is the first time a brain derived neuropeptide drives circadian oscillation in a peripheral organ and suggest NPY may be targeted to improve circadian regulation of xenobiotic metabolism.

Most of the experiments are straightforward in which the authors test mRNA expression of candidate gene by qPCR in various fly lines or liver gene expression in NPY knockout mice.

Given the role of NPY and NPF in feeding, and feeding is known to drive ~24 hr oscillation in gene expression in the complete absence of a clock anywhere in the body, it is important to have some measure of feeding pattern in these experiments in both flies and mice.

As noted by Reviewer 1, feeding rhythms are relatively low amplitude and so it would be difficult to see changes in these. We did examine the response of *sxe1* to restricted feeding and found that it is not affected any more than *per* (see response to Reviewer 2, and Figure 7). In addition, as noted above, the effect of NPF on *sxe1* is developmental and, therefore, not a direct response to feeding. Regarding *Cyp6a21*, we showed previously that while it can be driven by restricted feeding (RF) in *Clk^jrk^*flies the rhythm is not as robust as seen upon RF in wild type flies (Xu et al., 2011). Thus, it does not appear to be driven by a feeding cycle alone. Finally, the Panda database (http://circadian.salk.edu/about.html) does not indicate restoration of *Cyp2b10* cycling by restricted feeding of clockless mice. While all these data do not exclude a role of feeding, they indicate that NPF/NPY likely does not act through feeding alone to drive expression of these peripheral genes. We note too that only a limited subset of circadian genes that are independent of the liver clock depend on NPY. If NPY were driving gene expression through rhythmic feeding alone, it would impact most liver genes. The potential role of feeding is now fully discussed in the Discussion section.

*In rodents, NPY is also expressed in peripheral tissues including the gut. A recent paper from Chang's lab has shown the gut plays a profound role in driving liver circadian gene expression. Therefore, it is premature to state the CNS expression of NPY is driving liver circadian gene expression. The authors may use a hypothalamus specific knockout of NPY if they want to drive this message. Along the same line, it will be worth examining whether NPF system is expressed in the peripheral organs of flies.* We acknowledge that the mouse data do not exclude a role for NPY function in the periphery. This possibility is now mentioned. However, in the fly, our data with the DvPdf driver show that LN_d_s are important for cycling of *sxe1*. In addition, the previous work of Jae Park and colleagues (2006) showed that in *Clk^jrk^*mutants *npf* is downregulated in LN_d_s, but not in other NPF-positive cells. While LN_d_s and NPF could act independently to regulate *sxe1*, it is more likely that LN_d_s are the source of NPF in this case.

With the lab's track record in genome wide studies and this paper being a follow-up of their earlier gene expression studies in fat body, it is worth including a RNA-seq or microarray study to examine to what extent the NPF system in flies and NPY system in mice regulate circadian fatbody/liver gene expression independent of feeding pattern. Such experiments will expand the impact and relevance of the paper beyond just a couple of liver transcripts.

We have done as the reviewer suggested, and examined the genome-wide impact of loss of NPY. The revised manuscript presents our new microarray data showing that the cycling of several liver transcripts is regulated by NPY.